# Analyzing the Effect of Noise in LLM Fine-tuning

## Abstract

Fine-tuning is the dominant paradigm for adapting pretrained large language models (LLMs) to downstream NLP tasks. In practice, fine-tuning datasets may contain various forms of noise arising from annotation errors, preprocessing artifacts, or automated data collection. Although prior work has focused on designing robust learning algorithms to mitigate performance degradation under noisy conditions, comparatively little is known about how different types of noise affect the internal learning dynamics of LLMs during fine-tuning. In this work, we systematically study the impact of noise on model behavior across five pretrained model families (GPT-2, Qwen2, Qwen-2.5-Instruct and Llama-2, Gemma3) and four diverse NLP tasks. We introduce controlled perturbations corresponding to three common real-world noise types: label noise, grammatical noise, and typographical noise. Beyond task-level performance, we analyze layer-wise representation changes and attention patterns to understand how noise propagates through the network. Our results show that the corruption of labels (i.e., label noise) consistently causes the largest performance degradation, whereas grammatical noise and typographical noise occasionally yield mild regularization benefits. We further find that effects of noise are localized primarily to task-specific layers, while attention structures remain comparatively stable. Our code is available here [1].

## 1 Introduction

Fine-tuning has become a dominant paradigm for adapting pretrained language models to downstream NLP tasks. Broadly speaking, fine-tuning implicitly assumes that the data used for training is reliable. In practice, training data is often noisy due to annotation errors, imperfect preprocessing pipelines, or automatic data collection methods such as web scraping and distant supervision Frenay & Verleysen (2014); Ratner et al. (2017); Zhang et al. (2025).

For instance, classification datasets may contain incorrect labels, while generation or understanding tasks frequently include grammatical errors, spelling mistakes. Although a substantial body of research has studied robust learning under adversarial or corrupted data Patrini et al. (2017); Han et al. (2018); Li et al. (2020), most of this work focuses on designing algorithms that mitigate performance degradation. Much less is understood about how different types of noise influence the internal learning dynamics of LLMs during fine-tuning.

Modern language models generally have a large number of parameters, and fine-tuning often modifies only a small subset of representations responsible for task-specific behavior. Consequently, different noise sources may affect distinct parts of the model in different ways, potentially leading to performance degradation or unexpected improvements.

In order to examine the above-mentioned hypothesis, we systematically investigated the effect of noise during fine-tuning across multiple model families and tasks. We explored five widely used pretrained models GPT-2, Qwen2, Qwen-2.5-Instruct, Llama-2 and Gemma3-1B-IT, and evaluated them on four diverse NLP tasks (i.e., **Sentiment Classification (SC)**, **Question Answering (QA)**, **Machine Translation (MT) and Math Reasoning (MR)**). Prior work (Subramaniam et al., 2009; Zhang et al., 2025) suggests that three categories of noise frequently arise in real-world text data. Consequently, we introduced controlled perturbations

---

[1] https://anonymous.4open.science/r/data-noise-influence-anonymous-383F

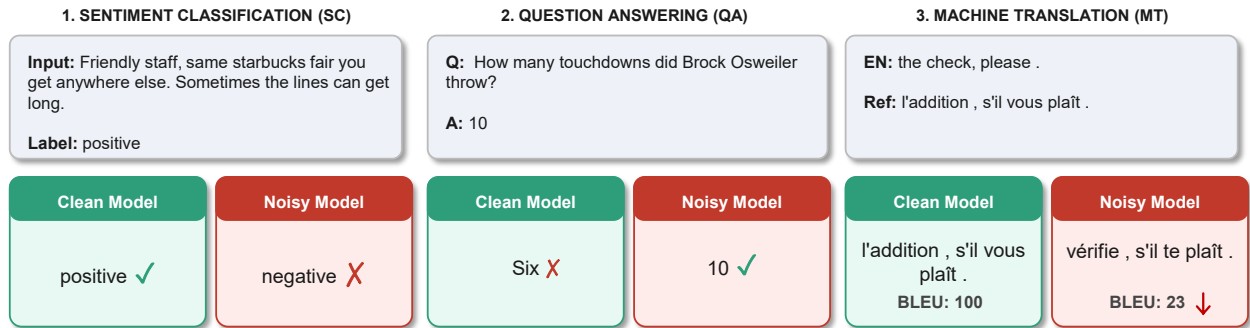

Figure 1: Examples of prediction changes in Llama-2 7B under different noise types at 40% corruption ratio: label-flip (SC), typographical (QA), and grammatical (MT).

corresponding to three different noise categories: *Label noise* Frenay & Verleysen (2014), *Typographical noise* Karpukhin et al. (2019), *Grammatical noise* Moradi & Samwald (2021). For all the above-mentioned noise types, we analyzed layer-wise representation changes and attention patterns to understand how noise propagates through the network. Our contributions can be summarized as follows.

- **Label noise is the most harmful:** Across all tasks and models, label noise resulted in the largest performance degradation, whereas grammatical and typographical noise occasionally caused marginal performance improvements.

- **Noise impact is localized:** Layers that encode higher levels of task-specific information generally exhibit greater distortion when noise is introduced.

- **Attention patterns remain stable:** Despite performance changes, token attention ordering changes only marginally.

Figure 1 illustrates the result of three different types of noise (i.e., label noise, typographical noise, grammatical noise) in different tasks (i.e., SC, QA, MT, and MR). As shown in Figure 1 that although label noise and grammatical noise caused incorrect predictions, typographical noise in fact improved the answer. The remainder of this paper is organized as follows. Section 2 reviews related work, Section 3 presents the methodology for analyzing the effects of noise, Section 4 describes the experimental setup, and Section 5 reports the experimental results. Section 6 further stratifies the evaluation samples into robust and vulnerable groups to examine whether the representational changes observed under noise are uniform across all test samples. Finally, Section 7 concludes the paper.

## 2  Related Work

Prior research on learning in noisy settings can be broadly grouped into two areas: (a) the effects of noise in fine-tuning and (b) robust learning with noisy labels.

### 2.1  Effect of Noise in Fine-tuning

Liu et al. (2020) identified the early-learning phenomenon which states neural networks learn clean patterns before memorizing noisy labels, motivating early stopping as a regularizer. Tänzer et al. (2022) further refined this concept in BERT fine-tuning on noisy NER data, identifying three temporal phases fitting, settling, and memorization and showed that noisy samples drift in embedding space during the memorization phase. Chen et al. (2025) characterises the fine-tuning loss landscape as nested basins, where adversarial fine-tuning can escape the stability region of the pretrained model. Ju et al. (2022) link flatter minima to greater noise resilience, and Li & Zhang (2021) provides PAC-Bayes bounds relating layer-wise weight distance to generalization under noise. Kim et al. (2024) found that parameter-efficient methods (LoRA,

adapters, prompt tuning) are generally more robust than full fine-tuning under label noise at rates of 20% to 60%, and attributed this to the low-rank bottleneck that limits memorization capacity.

The above-mentioned works primarily analyze noisy learning from parameter (Kim et al., 2024) and loss-level perspectives (Ju et al., 2022; Chen et al., 2025), typically focusing on classification tasks (Tänzer et al., 2022) and a single type of noise. In contrast, our work provides a complementary perspective by examining three types of noise at the representation level across widely used NLP tasks.

## 2.2 Robust Learning with Noisy Labels

Classical approaches to robust learning under label noise can be broadly grouped into four categories. Loss correction methods model the noise transition matrix to correct the training objective Patrini et al. (2017). Robust loss design replaces cross-entropy with noise-tolerant alternatives such as MAE or generalized cross-entropy Ghosh et al. (2017); Zhang & Sabuncu (2018). Sample selection methods leverage the memorization effect of deep networks, where clean samples consistently incur smaller losses early in training, to filter out unreliable examples Han et al. (2018). More recent work combines sample selection with semi-supervised learning, treating noisy-labelled instances as unlabelled data and jointly optimizing over both subsets Li et al. (2020). While effective for image classification, these methods focus on mitigating performance degradation, leaving largely unexplored the question of how different noise types reshape internal representations during fine-tuning. In a safety context, Rosati et al. (2024) showed that harmful fine-tuning recovers latent harmful representations rather than creating new ones, consistent with the wrapper view.

Recently, with the advent of pretrained language models there has been a large body of work for training with noisy labels. Zhu et al. (2022) found that BERT is robust to synthetic label noise but degrades substantially under realistic, instance-dependent noise. Wang et al. (2023) leveraged ChatGPT-generated rationales to separate clean from noisy samples during LLM fine-tuning. Their framework, LAFT, uses the agreement between the original noisy label and the LLM-predicted label as a confidence signal to partition training data into clean, ambiguous, and noisy subsets, each receiving different training strategies. Luo et al. (2024) extends noise-robust training to open-ended generation, moving beyond the classification setting. Most relevant to our experimental design, Zhu et al. (2024) shows that in machine translation fine-tuning, target-side noise (corrupted references) is substantially more damaging than source-side noise (corrupted inputs) — a finding that directly parallels our comparison of label corruption versus input-side (typographical, grammatical) noise. Similarly, Qi et al. (2024) demonstrates that as few as 10 adversarial examples can compromise safety alignment during fine-tuning. These studies focus on developing methods to resist noise or on measuring its impact on output performance. Our work complements this line of research by asking a different question: not how to maintain accuracy under noise, but how noise reshapes the internal representations of model.

# 3 Methodology for Noise Analysis

A model that exhibits similar, higher or lower task performance after being trained on noisy data may achieve this either by preserving its original task-specific representations (encoding obtained from training on clean data) or by learning fundamentally different encoding strategies. Standard task-level metrics (e.g., accuracy, BLEU) cannot distinguish between these two scenarios. To disentangle these possibilities, we employ three complementary analysis methods that examine (i) how noise alters attention patterns, (ii) how noise affects task-relevant information encoded within the model, and (iii) how internal representations change before and after fine-tuning.

Throughout this section, we compare a clean model (fine-tuned on unperturbed data) with a noisy model (fine-tuned on corrupted data); noisy-model quantities are distinguished by a tilde. Subscripts index the layer $\ell$ and sample $s$; $\mathcal{S}$ denotes the clean held-out test set.

Table 1: Summary of different metrics used to analyze the effect of noise. All metrics are computed at every layer $\ell$.

| Metric | Analysis Aspect |
|---|---|
| $\overline{D}_{KL}(\ell)$ | Attention value divergence |
| $\bar{\rho}_k(\ell)$ | Attention priority order |
| $\mathcal{M}_\ell$ | Task-aligned information |
| $\mathrm{Cos}_{s,\ell}$ | Per-sample directional shift |
| $\mathrm{CKA}(\mathbf{H}_\ell, \tilde{\mathbf{H}}_\ell)$ | Inter-sample structural change |

### 3.1 Attention Matrix Analysis

In order to examine whether noise alters the attention pattern, we conduct two complementary analyses. The first focuses on a) *attention values* and the second on b) *the order of tokens* (i.e., tokens having the highest attention to the lowest attention).

To understand the change in values within attention matrices, we compute the Kullback-Leibler (KL) divergence between the clean and noisy attention distributions, averaged over all clean test samples, attention heads, and token positions at each layer. Let $\mathbf{a}_{h,t}^{(s)}$ and $\tilde{\mathbf{a}}_{h,t}^{(s)}$ denote the clean and noisy attention weight vectors for head $h$, token position $t$, and sample $s$, with $H$ heads per layer and $T_s$ tokens per sample. Formally,

$$\overline{D}_{\mathrm{KL}}(\ell) \;=\; \frac{1}{|\mathcal{S}|} \sum_{s \in \mathcal{S}} \frac{1}{H} \sum_{h=1}^{H} \frac{1}{T_s} \sum_{t=1}^{T_s} D_{\mathrm{KL}}\big(\mathbf{a}_{h,t}^{(s)} \,\|\, \tilde{\mathbf{a}}_{h,t}^{(s)}\big) \tag{1}$$

A high $\overline{D}_{\mathrm{KL}}(\ell)$ indicates that noise has substantially altered how layer $\ell$ distributes contextual importance across tokens.

While KL divergence captures changes in attention magnitude, it does not reveal how much the order of important tokens in a particular context has changed due to noise. To investigate this phenomenon, we also compute Spearman rank correlation coefficient between clean and noisy attention weights , restricted to the top-$k$ tokens for each token position ($t$), and averaged in the same manner as in to Equation 1. Formally,

$$\bar{\rho}_k(\ell) \;=\; \frac{1}{|\mathcal{S}|} \sum_{s \in \mathcal{S}} \frac{1}{H} \sum_{h=1}^{H} \frac{1}{T_s} \sum_{t=1}^{T_s} \rho\big(\mathbf{a}_{h,t}^{(s)}, \ \tilde{\mathbf{a}}_{h,t}^{(s)}\big) \tag{2}$$

High $\bar{\rho}_k$ together with high KL indicates attention value redistribution without a change in priority, whereas a high KL divergence together with a low $\bar{\rho}_k$ indicates a major reordering of attention targets.

For attention pattern analysis, we examine the attention patterns on the same set of clean inputs for both the clean and the noisy model. Consequently, there is no difference in the number of tokens between the two models. The metrics used for attention matrix analysis complement the hidden-state analyses (outlined in subsection 3.2 and subsection 3.3) by revealing whether representational changes originate from altered attention patterns or from transformations within the feed-forward sublayers.

### 3.2 Probing

To assess whether task-relevant information remains encoded within different layers of LLMs trained on noisy data in a manner similar to the clean model, we probe every layer with the Logit Lens (nostalgebraist, 2020). Let $\mathcal{S}$ denote the clean test set, where each sample $s \in \mathcal{S}$ consists of an input prompt $x^s$ and a ground-truth target sequence $y^s = (y_1^s, \ldots, y_{T_s}^s)$ of length $T_s$. Predictions are obtained by teacher forcing (Goodfellow et al., 2016): to obtain the prediction at target position $j$, the model is conditioned on the prompt $x^s$ together with the *ground-truth* prefix $y_{1:j-1}^s = (y_1^s, \ldots, y_{j-1}^s)$, i.e., the true preceding tokens rather than the past predictions of the model. We then apply the Logit Lens at each layer $\ell$: the hidden state at layer $\ell$ and position $j$ is

Table 2: Supervised fine-tuning task-related performance of different models under different noise ratios (SC: Accuracy; QA: F1 score; MT: BLEU score; MR: Exact Match). All noisy results are compared against the corresponding **Clean FT** baseline within each row: cells that degrade relative to Clean FT are shaded in red , and cells that improve or equal to are shaded in green . For each row, the **bold** cell marks the largest performance drop relative to Clean FT.

| | | Baselines | | Label Flip | | | Typo Error | | | Gramm. Error | | |
|---|---|---|---|---|---|---|---|---|---|---|---|---|
| Task | Model | Pretrained | Clean FT | 20% | 30% | 40% | 20% | 30% | 40% | 20% | 30% | 40% |
| | | | | | | QLoRA | | | | | | |
| SC | GPT-2 | 0.12 | 92.30 ± 0.26 | 91.07 ± 0.26 | 87.93 ±1.67 | **73.93 ± 1.11** | 91.77 ± 0.09 | 92.17 ± 0.61 | 91.83 ± 0.68 | 91.87 ± 0.34 | 92.00 ± 0.50 | 91.83 ± 0.31 |
| | Qwen2 | 16.30 | 94.47 ± 0.12 | 90.37 ± 0.54 | 79.07 ± 2.90 | **60.27 ± 3.28** | 94.20 ± 0.16 | 94.07 ± 0.05 | 94.33 ± 0.12 | 94.30 ± 0.22 | 94.40 ± 0.14 | 94.33 ± 0.12 |
| | Llama-2 | 0.60 | 94.40 ±0.26 | 94.43 ± 2.04 | 89.97 ± 8.71 | **62.03 ± 31.50** | 94.30 ± 0.36 | 94.40 ± 0.00 | 94.27 ± 0.12 | 94.37 ± 0.12 | 94.10 ± 0.08 | 94.17 ± 0.29 |
| | Qwen2.5-1.5B-Instruct | 92.70 | 94.44 ±0.15 | 94.33 ± 0.17 | 93.83 ± 0.09 | **89.63 ± 0.52** | 94.47 ± 0.12 | 94.20 ± 0.28 | 94.30 ± 0.14 | 94.47 ± 0.05 | 94.50 ± 0.00 | 94.50 ± 0.08 |
| | Gemma3-1B-IT | 90.60 | 93.37 ±0.04 | 93.41 ± 0.06 | 91.78 ± 0.12 | **88.41 ± 0.08** | 93.67 ± 0.11 | 93.43 ± 0.07 | 93.85 ± 0.12 | 93.77 ± 0.04 | 93.85 ± 0.02 | 93.71 ± 0.03 |
| QA | GPT-2 | 8.11 | 55.04 ± 0.78 | 51.87 ± 1.21 | 50.43 ± 0.75 | **49.38 ± 0.76** | 53.89 ± 1.05 | 54.26 ± 0.85 | 52.82 ± 0.65 | 55.20 ± 1.11 | 53.51 ± 0.85 | 54.51 ± 1.05 |
| | Qwen2 | 38.75 | 68.85 ± 1.06 | 61.32 ± 0.91 | 52.13 ± 0.75 | **49.65 ± 0.45** | 70.21 ± 1.10 | 69.81 ± 0.73 | 69.88 ± 0.82 | 71.20 ± 1.09 | 70.88 ± 0.79 | 69.88 ± 1.03 |
| | Llama-2 | 30.90 | 85.91 ± 0.65 | 78.87 ± 1.03 | **51.95 ± 1.06** | 74.12 ± 0.91 | 86.32 ± 0.73 | 86.44 ± 0.35 | 82.98 ± 1.12 | 85.41 ± 0.72 | 85.73 ± 1.14 | 86.33 ± 0.46 |
| | Qwen2.5-1.5B-Instruct | 53.80 | 87.12 ± 1.11 | 85.89 ± 0.82 | 81.92 ± 0.95 | **80.99 ± 0.91** | 86.33 ± 0.69 | 85.48 ± 0.54 | 84.89 ± 1.05 | 86.52 ± 0.61 | 86.55 ± 0.71 | 85.96 ± 1.18 |
| | Gemma3-1B-IT | 53.93 | 66.65 ± 0.11 | 51.38 ± 0.12 | 66.31 ± 0.05 | **62.77 ± 0.09** | 63.89 ± 0.17 | 65.11 ± 0.06 | 65.33 ± 0.24 | 67.06 ± 0.05 | 66.62 ± 0.11 | 67.51 ± 0.04 |
| MT | GPT-2 | 0.31 | 21.52 ± 0.86 | 18.71 ± 0.46 | 15.81 ± 0.73 | **14.56 ± 1.03** | 21.05 ±0.69 | 21.19 ± 0.77 | 21.67 ± 0.31 | 22.91 ± 0.52 | 22.56 ± 0.49 | 22.39 ± 0.27 |
| | Qwen2 | 14.49 | 43.95 ± 1.04 | 43.22 ± 0.12 | 42.78 ± 0.47 | **42.61 ± 0.13** | 45.71 ± 0.62 | 45.18 ± 0.24 | 44.88 ± 0.71 | 44.11 ± 0.12 | 45.12 ± 0.89 | 44.38 ± 0.25 |
| | Llama-2 | 20.67 | 52.03 ± 0.18 | 51.56 ± 0.35 | **49.75 ± 0.25** | 50.37 ± 0.67 | 53.15 ± 0.23 | 51.82 ± 0.11 | 51.84 ± 0.35 | 52.17 ± 0.12 | 51.77 ± 0.11 | 52.46 ± 0.23 |
| | Qwen2.5-1.5B-Instruct | 18.87 | 50.46 ± 0.23 | 48.75 ± 0.15 | **43.13 ± 0.33** | 47.85 ± 0.55 | 50.41 ± 0.33 | 50.47 ± 0.27 | 49.79 ± 1.03 | 50.56 ± 0.26 | 50.82 ± 0.13 | 50.68 ± 0.28 |
| | Gemma3-1B-IT | 21.72 | 53.69 ± 0.35 | 51.38 ± 0.12 | **50.85 ± 0.21** | 50.91 ± 0.07 | 52.95 ± 0.13 | 52.33 ± 0.14 | 52.51 ± 0.11 | 53.25 ± 0.11 | 52.48 ± 0.12 | 52.31 ± 0.15 |
| MR | GPT-2 | 1.20 | 1.87 ± 0.21 | 1.70 ± 0.24 | 1.80 ± 0.37 | 1.83 ± 0.24 | 1.87 ± 0.26 | 1.80 ± 0.29 | 1.70 ± 0.43 | **1.60 ± 0.36** | 1.63 ± 0.45 | 1.67 ± 0.42 |
| | Qwen2 | 10.10 | 40.63 ± 0.76 | 40.10 ± 0.54 | 39.70 ± 0.43 | 39.43 ± 1.18 | **39.07 ± 0.47** | 40.63 ± 0.87 | 39.50 ± 0.28 | 40.80 ± 0.43 | 40.43 ± 0.31 | 40.63 ± 0.88 |
| | Llama-2 | 5.20 | 53.63 ± 1.12 | **50.97 ± 1.07** | 51.40 ± 1.18 | 51.60 ± 0.29 | 53.33 ± 0.96 | 52.13 ± 0.45 | 53.10 ± 0.29 | 52.73 ± 0.24 | 51.93 ± 0.52 | 53.07 ± 0.38 |
| | Qwen2.5-1.5B-Instruct | 19.60 | 72.03 ± 0.12 | 70.60 ±0.28 | 70.87 ±0.29 | **70.37 ±0.29** | 71.23 ±0.37 | 71.27 ±0.05 | 71.03 ± 0.26 | 71.63 ± 0.12 | 72.20 ± 0.08 | 72.33 ± 0.12 |
| | Gemma3-1B-IT | 28.05 | 44.89 ± 0.56 | 43.82 ± 0.32 | **43.45 ± 0.12** | 43.85 ± 0.43 | 45.65 ± 0.14 | 44.62 ± 0.23 | 44.73 ± 0.06 | 46.05 ± 0.03 | 46.65 ± 0.11 | 45.82 ± 0.08 |

mapped to the vocabulary space using the same output prediction head as the final-layer representation, and the per-layer, per-position prediction is read out as

$$\hat{y}_{\ell,j}^{(s)} \;=\; \arg\max \; p_\ell\big(\cdot \mid x^s, \, y_{1:j-1}^s\big), \tag{3}$$

where $p_\ell(\cdot \mid \cdot)$ denotes the next-token distribution obtained by applying the final-layer output head (the unembedding) to the layer-$\ell$ hidden state. Collecting these over all target positions yields the per-layer prediction $\hat{y}_\ell^{(s)} = (\hat{y}_{\ell,1}^{(s)}, \dots, \hat{y}_{\ell,T_s}^{(s)})$. The probing procedure is identical across tasks; what differs is the metric $\mathcal{M}$ used to compare $\hat{y}_\ell^{(s)}$ against the ground truth $y^s$:

$$\mathcal{M}_\ell \;=\; \frac{1}{|\mathcal{S}|} \sum_{s \in \mathcal{S}} \mathcal{M}(\hat{y}_\ell^{(s)}, \, y^s). \tag{4}$$

Equation 4 formulates the metric $\mathcal{M}_\ell$ used to measure how useful the representation at layer $\ell$ is for a given task. We instantiate $\mathcal{M}_\ell$ as top-1 token accuracy for sentiment classification, SQuAD F1 (Rajpurkar et al., 2016) for question answering, BERTScore (Zhang et al., 2020) for machine translation, and numeric exact match for mathematical reasoning. A higher $\mathcal{M}_\ell$ indicates that the representation at layer $\ell$ encodes more of the information needed to estimate the correct output, which allow us to compare how task-aligned information is distributed across layers in the clean and noisy models.

### 3.3 Similarity Based on Input Representations

To further quantify the change in input representations under noise, two different similarity measures were computed. The first one is the centered cosine similarity between the hidden states of the clean and noise-trained models, computed for each evaluation sample at each layer (i.e., $\mathrm{Cos}_{s,\ell}$). Both the mean and standard deviation across all $|\mathcal{S}|$ evaluation samples are reported.

While cosine similarity measures individual vector alignment, it is insensitive to changes in encoding layer patterns across samples. To capture such changes, Linear Centered Kernel Alignment (CKA)(Kornblith et al., 2019) is computed between the representations of clean and noisy models.

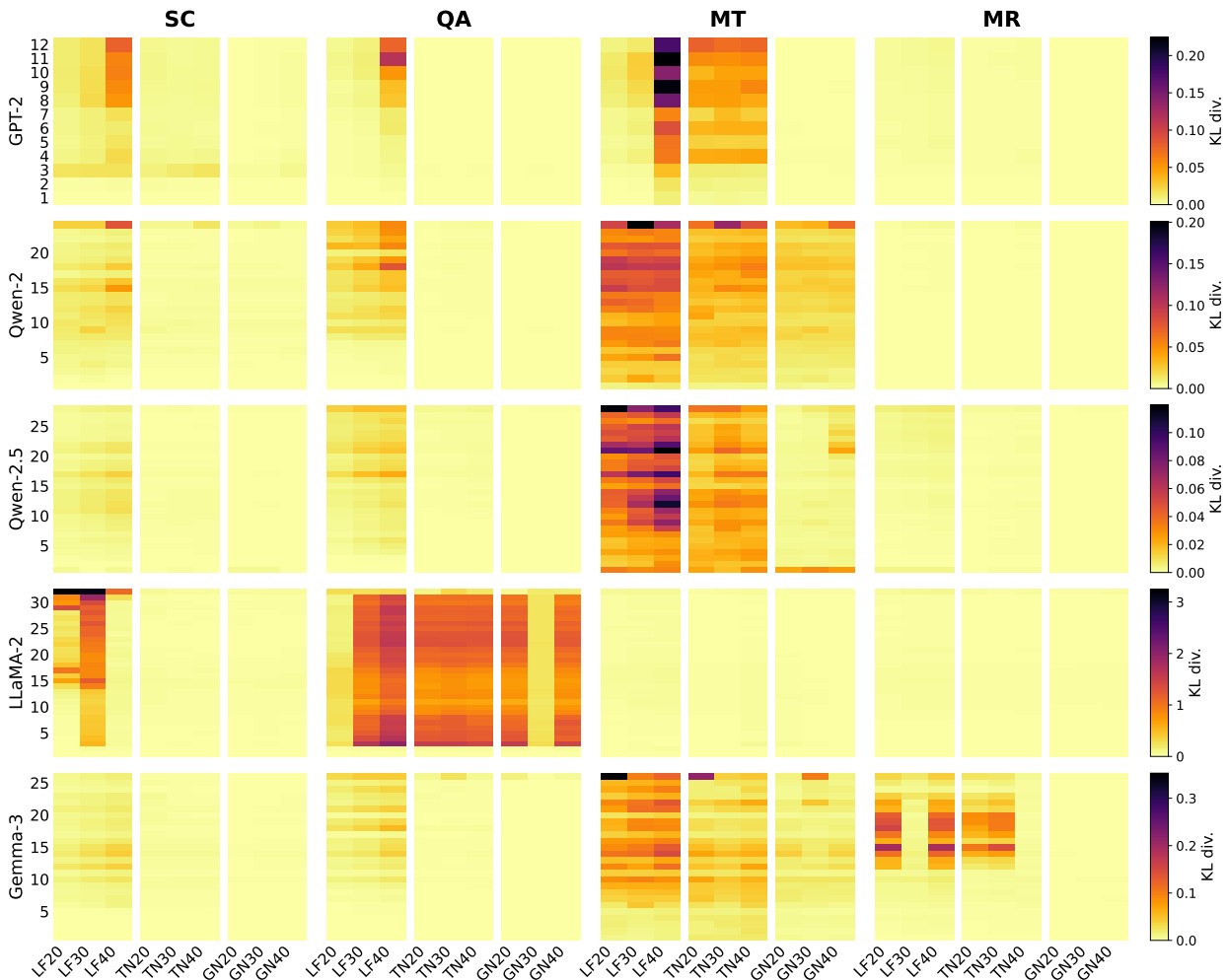

Figure 2: Layer-wise attention pattern divergence (KL divergence) between clean and noise-trained models. Rows correspond to GPT-2, Qwen-2, Qwen-2.5-Instruct, Llama-2 and Gemma3-1B-IT; columns correspond to SC, QA, MT and MR tasks. The x-axis denotes noise type and corruption ratio; the y-axis indicates the layer index of each model. Each cell shows the KL divergence averaged across all attention heads at a given layer. Each row uses an independent color scale because of differing divergence magnitudes across architectures.

For $|\mathcal{S}|$ evaluation samples, let $\mathbf{H}_\ell \in \mathbb{R}^{|S| \times d}$ and $\tilde{\mathbf{H}}_\ell \in \mathbb{R}^{|S| \times d}$ denote the centered hidden-state matrices from the clean and noise-trained models at a given layer $\ell$, respectively. Linear CKA is defined as

$$\mathrm{CKA}(\mathbf{H}_\ell, \tilde{\mathbf{H}}_\ell) \;=\; \frac{\left\|\tilde{\mathbf{H}}_\ell^\top \mathbf{H}_\ell\right\|_F^2}{\left\|\mathbf{H}_\ell^\top \mathbf{H}_\ell\right\|_F \;\cdot\; \left\|\tilde{\mathbf{H}}_\ell^\top \tilde{\mathbf{H}}_\ell\right\|_F} \tag{5}$$

where $\|\cdot\|_F$ denotes the Frobenius norm (Golub & Van Loan, 2013). Crucially, CKA is invariant to orthogonal transformations and isotropic scaling. If training on noise merely rotated the representation space without altering its internal geometry, CKA would remain near 1.0 regardless of how low the cosine similarity drops. Conversely, a low CKA value provides evidence that the inter-sample relational structure has been fundamentally altered, a change that cosine similarity alone cannot detect.

Table 1 summarizes the metrics discussed above in analyzing the effects of noise across four tasks and five models.

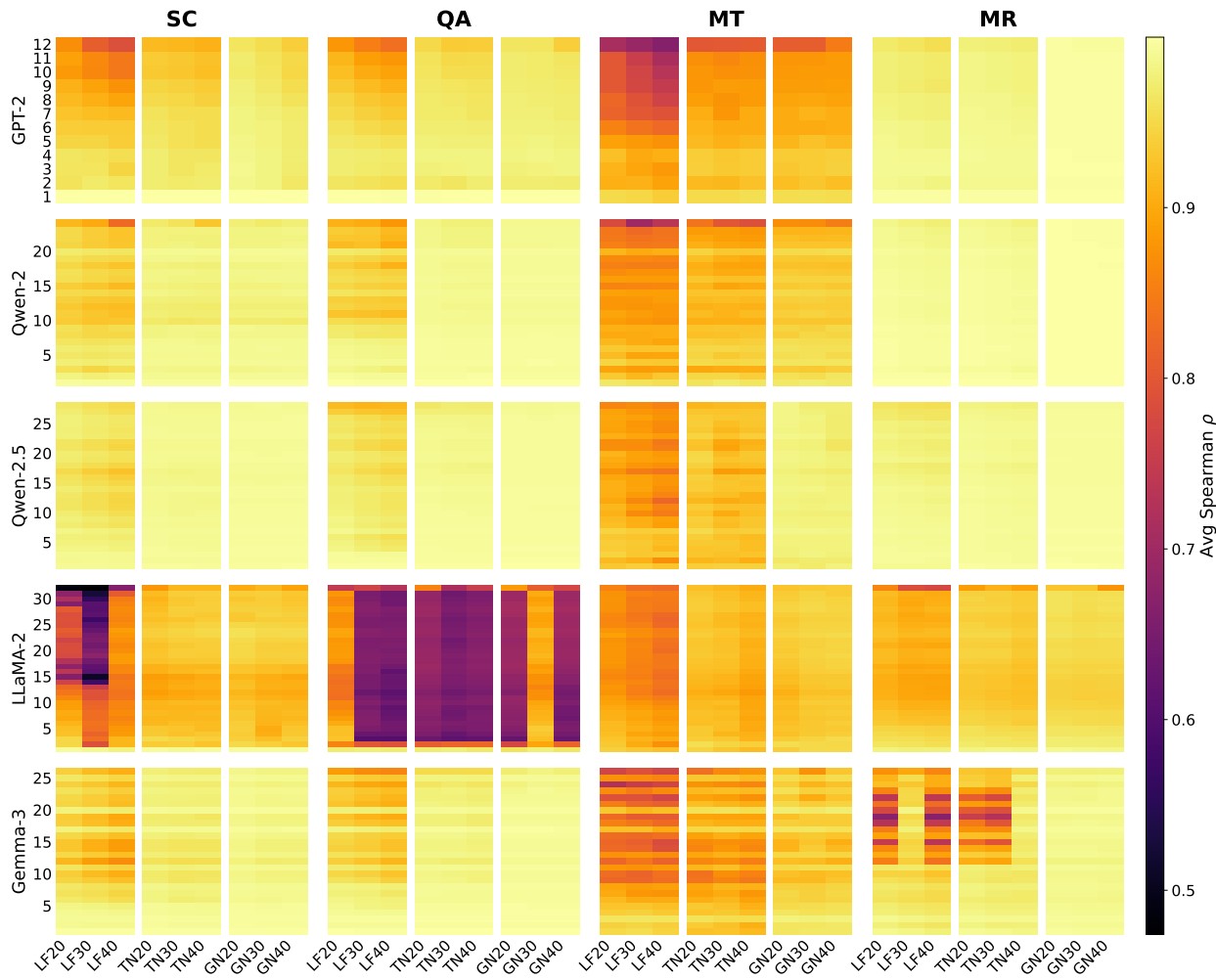

Figure 3: Layer-wise attention pattern stability measured by Spearman rank correlation ($\rho$) between clean and noise-trained models. Rows correspond to GPT-2, Qwen-2, Qwen-2.5-Instruct, Llama-2 and Gemma3-1B-IT; columns correspond to SC, QA, MT and MR tasks. The x-axis denotes noise type and corruption ratio; the y-axis indicates the layer index of each model.

# 4 Experimental Setup

In this section, we describe the dataset setup, followed by the noise incorporation mechanism, fine-tuning setup, and evaluation metrics.

**Tasks & Datasets.** As described in Section 1, we explore four different NLP tasks: a) Sentiment Classification (SC), b) Question Answering (QA), and c) Machine Translation (MT) and Math Reasoning (MR), all of which are framed as generative tasks. For SC, we consider binary sentiment classification on movie reviews from the Yelp Polarity dataset (Zhang et al., 2015). For QA the task is given a passage and a question, and the relevant portion is extracted from the passage as the answer. We use SQuAD v1.1 (Rajpurkar et al., 2016) dataset for this task. For MT, we focus on English-to-French translation from Tatoeba parallel corpus (Tiedemann, 2020). For MR, the task is to read a grade-school math word problem and generate a step-by-step solution ending with the final numeric answer. We use MetaMathQA (Yu et al., 2024) for fine-tuning and evaluate on the GSM8K (Cobbe et al., 2021) test set. Further dataset details (e.g., sample example from the dataset) are provided in Table 3 (Appendix A).

**Noise Types.** We explore three different noise types in our experiment setup a) *Label Flip* (LF), b) *Typographical Noise* (TN) and c) *Grammatical Noise* (GN). Existing research (Subramaniam et al., 2009; Bryant et al., 2022) shows that the above-mentioned three different types of noise primarily cover most of the widely observed noise in NLP tasks. Each one of them is described as follows.

- **LF** Here, only the target output is corrupted. For SC, this flips the polarity label (positive ↔ negative). For QA, the gold answer is replaced with a randomly sampled answer from the training pool, and for MT the reference translation is replaced with an unrelated target sentence. For MR, the entire chain-of-thought response is replaced with one generated by DeepSeek-Math-7B-Instruct on the same question and retained only if its final numerical answer disagrees with the ground truth, following the incorrect-answer reasoning-path setting in Yu et al. (2024).

- **TN** Here, character-level perturbations (e.g., deletion, swap, insertion, or substitution of a single character) to approximately 10% of words in the *input* text (review, context, or source sentence), are applied to simulate common typing errors similar to Gao et al. (2018).

- **GN** Here, rule-based substitutions targeting subject–verb agreement (*is↔are*, *was↔were*, *has↔have*) and article misuse (e.g., *an apple → a apple*) into the input text at a rate of approximately 15% word, similar to Moradi & Samwald (2021).

Prior research shows that classical noise-robust learning algorithms require the rate of noise to remain below 0.5 (Angluin & Laird, 1988; Natarajan et al., 2013). Following prior study in noisy text classification (Liu et al., 2022), which find that many methods degrade significantly beyond 30% noise, we evaluate three noise levels: 20%, 30%, and 40%, covering the range from moderate to near-critical noise conditions. Examples of the above-mentioned types of noise are given in Appendix A.1.

**Fine-tuning Setup.** We fine-tune different LLMs (i.e., GPT-2 124M Radford et al., 2019, Qwen2-0.5B Yang et al., 2024, Qwen-2.5-1.5B-Instruct Yang et al., 2024, Llama-2-7B Touvron et al., 2023, Gemma3-1B-IT Kamath et al., 2025) in our experiment setup. Specifically, Qwen-2.5-1.5B-Instruct and Gemma3-1B-IT operate instruction fine-tuning through all the tasks. Details of fine-tuning prompts are in Table 4 of Appendix. All five models are fine-tuned with QLoRA (Dettmers et al., 2023) using 4-bit NF4 quantization, giving a consistent fine-tuning paradigm across model scales. For all models, LoRA adapters are applied to the attention and feed-forward projection layers; the per-task rank $r$, scaling factor $\alpha$, and dropout are given in Table 9(Appendix B), and the remaining training hyperparameters in Table 10(Appendix B).

All models are trained using the SFT-MASK protocol implemented via the `SFTTrainer` from the TRL library (von Werra et al., 2020), the cross-entropy loss is computed only over completion tokens, with prompt tokens excluded from the loss computation. To verify the stability of our findings, we perform a similar seed-based analysis for CKA to investigate seed-induced representational variance (Appendix H). Implementation details are provided in Appendix B.

**Evaluation.** For each task, we use a task-specific metric to evaluate the overall performance on the task. For SC we use accuracy (whether the generated completion matches the gold label), for QA we use token-level SQuAD F1, and for MT we use BLEU score (computed with SacreBLEU Post, 2018) , and for **MR** we use exact-match (EM) accuracy on the final numeric answer of each GSM8K problem. Each model–task pair is evaluated on the same clean held-out test set of examples (Table 3).Noise is injected only into the training data; the evaluation inputs are never perturbed. All models and tasks are evaluated with greedy decoding; the full per-task generation settings (generation length and the repetition penalty applied for QA and MT) are reported in Appendix C.

**Layer-wise analysis metric.** For the teacher-forced logit-lens analysis (Figure 4) we measure per-layer task information with the same metrics as above, except that MT uses BERTScore F1 rather than corpus BLEU: decoding a single layer's hidden state yields very short, frequently empty hypotheses for which corpus-level BLEU collapses to zero and is uninformative, whereas BERTScore provides a smooth, semantically meaningful per-layer signal. Overall task performance (Table 2) continues to use BLEU for MT.

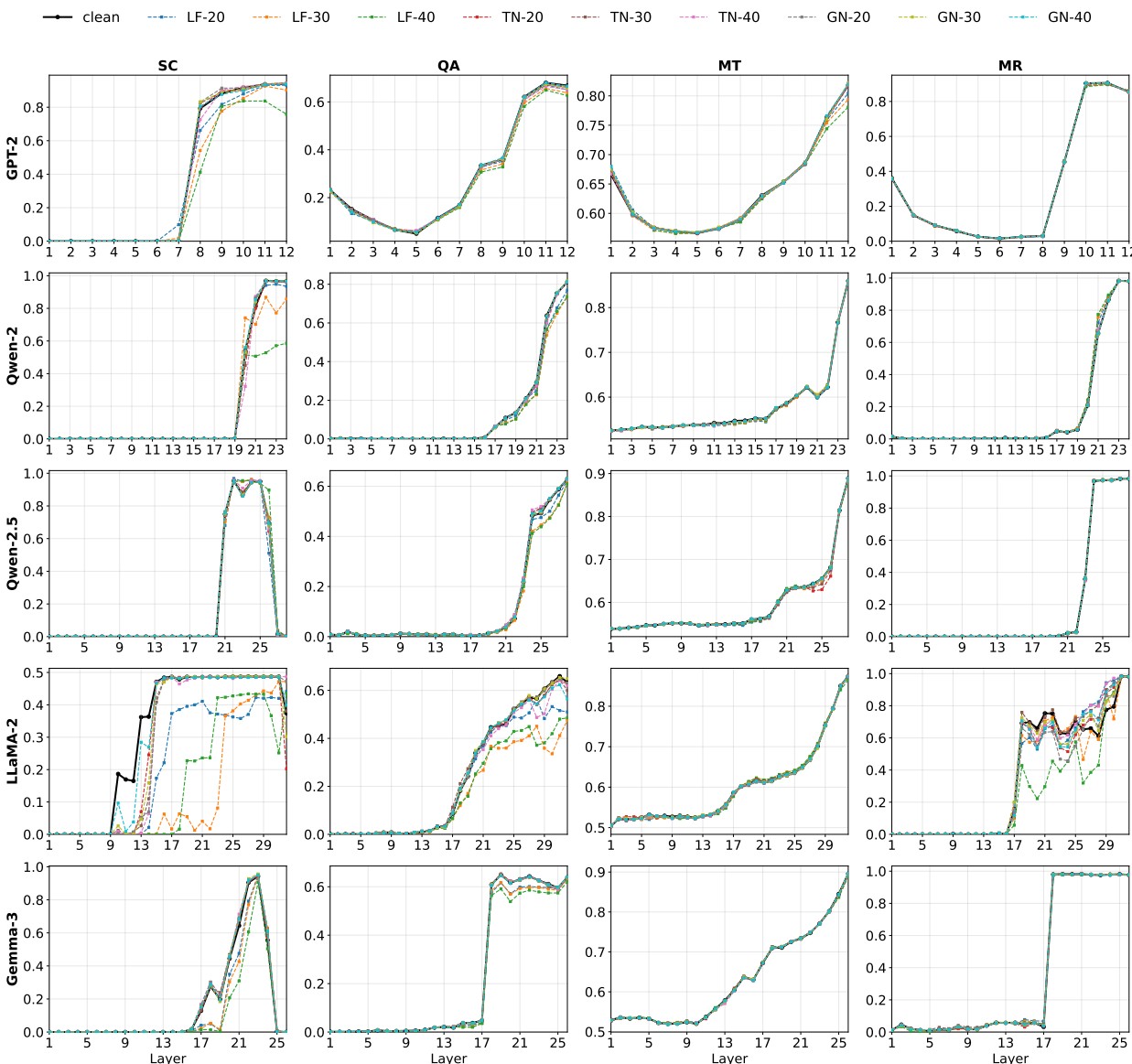

Figure 4: Layer-wise task-information analysis (teacher-forced logit lens) for QLoRA-fine-tuned models. Each row is a model (GPT-2, Qwen2, Qwen-2.5-Instruct, Llama-2 and Gemma3-1B-IT) and each column is a task (SC, QA, MT, MR). Within every panel the x-axis is the layer index and the y-axis is that layer's task performance under the teacher-forced logit lens, which is decoding each layer's hidden state through the unembedding and scoring it with the task metric: SC = top-1 token accuracy, QA = SQuAD F1, MT = BERTScore F1, MR (math reasoning) = numeric exact-match on the extracted answer. The black curve is the clean-trained model; the nine colored curves are the noise-trained variants — LF = label-flip, TN = typo, GN = grammatical, each at 20/30/40% corruption.

# 5 Results

## 5.1 Task Performance Under Noise

Table 2 displays the model performance in different tasks under different types of noise, the main findings from it are as follows: **a)** For all types of tasks and across all the models, LF noise (40%) has generally caused the most amount of damage in most cases. In general, LF caused more damage compared to other forms of

noise. **b)** Input-side noise (i.e., typo noise and grammatical noise) is far less harmful than label noise (i.e., where noise was introduced in target), and can be mildly beneficial. In contrast to LF, both TN and GN leave task performance close to the clean baseline (Table 2): TN causes no large drops and in a few QA and MT settings marginally exceeds the clean model, staying within the margin of noise elsewhere. This pattern is consistent with the long-standing result that injecting small input perturbations acts as a regularizer improving generalization (Bishop, 1995), and with empirical evidence that training on synthetic textual noise improves robustness to natural noise (Karpukhin et al., 2019). The asymmetry between input-side (TN/GN) and target-side (LF) corruption further echoes Zhu et al. (2024), who report that target-side noise is substantially more damaging than source-side noise in MT fine-tuning. **c)** For SC and MT tasks, noise has affected the smaller models more compared to the larger models. However, for QA models, the same amount of noise has affected the larger model more compared to the smaller models. Notably, the two instruction-tuned models (Qwen-2.5-Instruct and Gemma3-1B-IT) are far more LF-robust on SC (about 5% drop) than the base models (18–36%), suggesting instruction-tuned initialization confers label-noise robustness.For the same task and similar types of noise, how the model will be affected depends on its architecture (e.g., First three rows in Table 2).

## 5.2 Attention Pattern Shifts

To further investigate the cause of damage by noise, we observed how attention patterns have changed across different models and different tasks in Figure 2. The primary observations from Figure 2 is as follows. **a)** Since LF noise generally is causing the most amount of damage, correspondingly, we see larger changes in attention patterns for LF. This is observed for all tasks and all models. **b)** Across all models the shifts concentrate in the deeper half of the network rather than the initial layers: the KL centroid lies beyond 0.5 of network depth in all 20 cells (range 0.52–0.81), and at most one third of the total shift mass falls in the first third of layers. For GPT-2 the per-layer KL rises almost monotonically from near zero at layer 0 to its maximum at the final layers; Qwen2-0.5B, Qwen2.5-1.5B-Instruct, and Llama-2 (on SC/MT/MR) likewise peak in their last layers. This concentration in later, task-specific layers is consistent with the representation-level findings in subsection 5.3. **c)** Another important thing we have observed is that the magnitude of the change is very small overall across all the models and all the task types. Broadly speaking, the attention matrices are not that susceptible to noise. **d)** With the increase in the amount of noise, the change is stronger no matter for better or worse performance. Observations from Figure 3 are very similar to Figure 2. It shows that order and magnitude of attention values follow the same pattern.

In Figure 7, Figure 8 and Figure 9, we identify the top-3 attention heads that change the most in response to different types of noise. We observe that, for each task and model, there is substantial overlap across noise types, with two out of the three most affected heads often being shared. This suggests that each model contains certain task-specific attention heads that are particularly sensitive to noise.

## 5.3 Layer-wise Task Information Under Noise

To further dig deeper into understanding the cause of damage due to noise, we observed probing accuracy for individual layers across all models and all tasks. Figure 4 shows the performance of probing. The key findings are as follows. **a)** Apart from Sentiment analysis there is mostly a consistent pattern in the performance of different layers across different models and different types of noise. The generic pattern is that the initial layers have a poor performance and then later layers are performing better showing that later layers have better task-performing ability compared to initial layers. **b)** Table 2 showed that the most damage was caused by label flip in sentiment analysis compared to any other configurations. This is visible from Figure 4 where we can see that the sentiment analysis label flip probing curve is maintaining the most distance from the clean model curve compared to other tasks. From Figure 4 we know that initial layers did not have any task-specific information. That's why the probing accuracies for initial layers are almost identical (approx to 0). This raised the question of whether the representations in the initial layers are also similar between clean and noisy models. Because two poor representations can yield similar poor downstream performance without providing any indication of how similar they are. In Figure 5 and Figure 13 (Appendix F), we have done centered cosine similarity / CKA similarity analysis across all the layers, and it can be observed that the

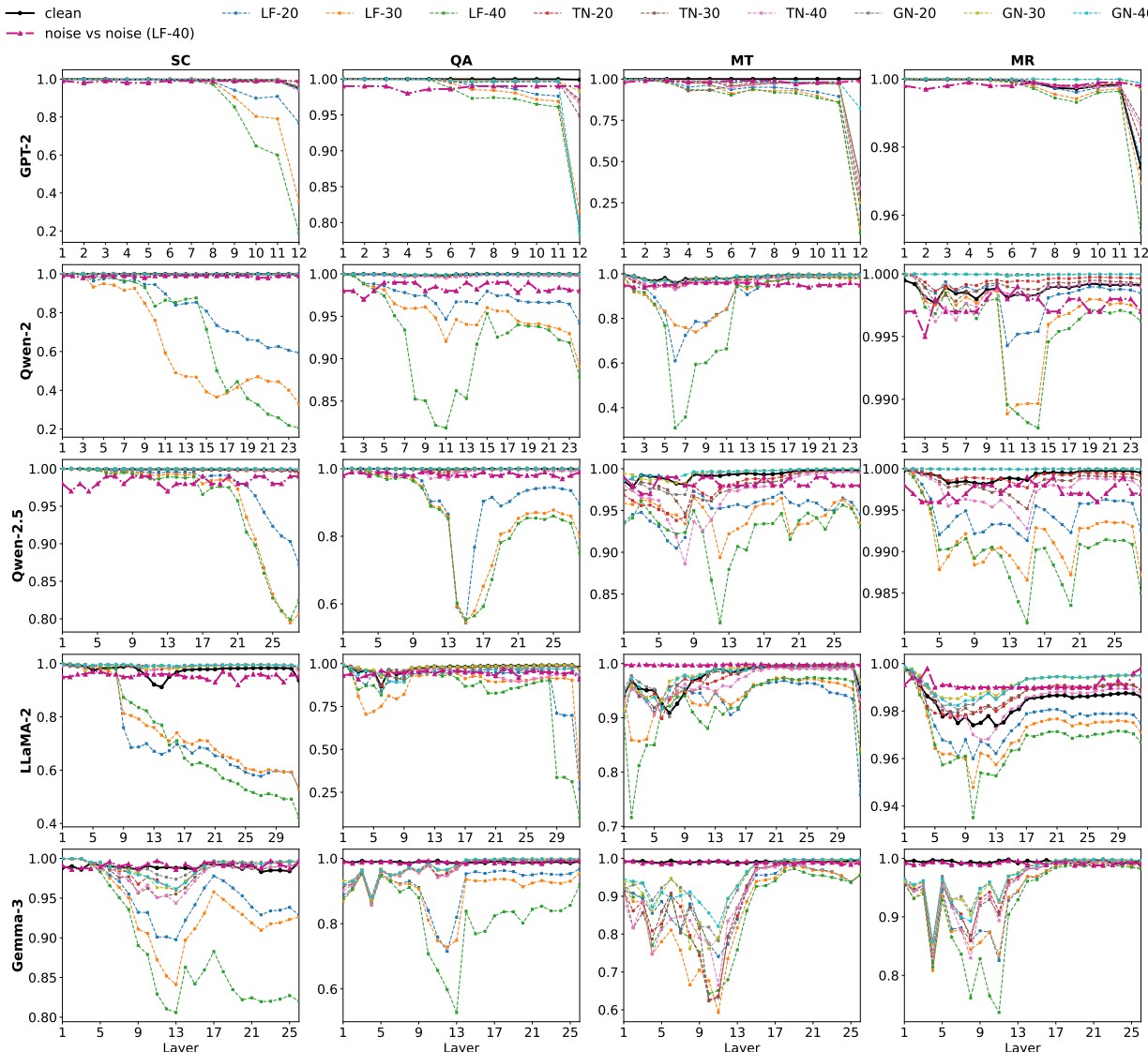

Figure 5: Layer-wise linear CKA similarity across four tasks (columns: SC, QA, MT, MR) and five models (rows). Row are GPT-2 Small (12 layers), Qwen2-0.5B (24 layers), Qwen2.5-1.5B-Instruct (28 layers) and Llama-2-7B (32 layers), and Gemma3-1B-IT (26 layers). Each panel shows three kinds of per-layer comparison: (i) the clean–clean cross-seed baseline ( average CKA between clean models trained with different seeds; black); (ii) clean vs. noise-trained models under nine corruption settings (label-flip, typo and grammatical noise at 20/30/40%; coloured); and (iii) the noise–noise cross-seed baseline (average CKA between LF-40 noise-trained models with different seeds; "noise vs noise (LF-40)"). The x-axis is the layer index and the y-axis is the linear CKA value (larger = more similar representations).

similarities are very high in the initial layers, indicating that task-specific noise primarily targets layers with more task information.

## 5.4  Representational Similarity

Figure 5 shows the layerwise linear CKA between the clean model and different types of noise. The main findings from Figure 5 are as follows. **a)** It can be observed that for most cases increasing more noise has created more distortion in the corresponding layer representation (e.g., for LF the green curves are mostly at

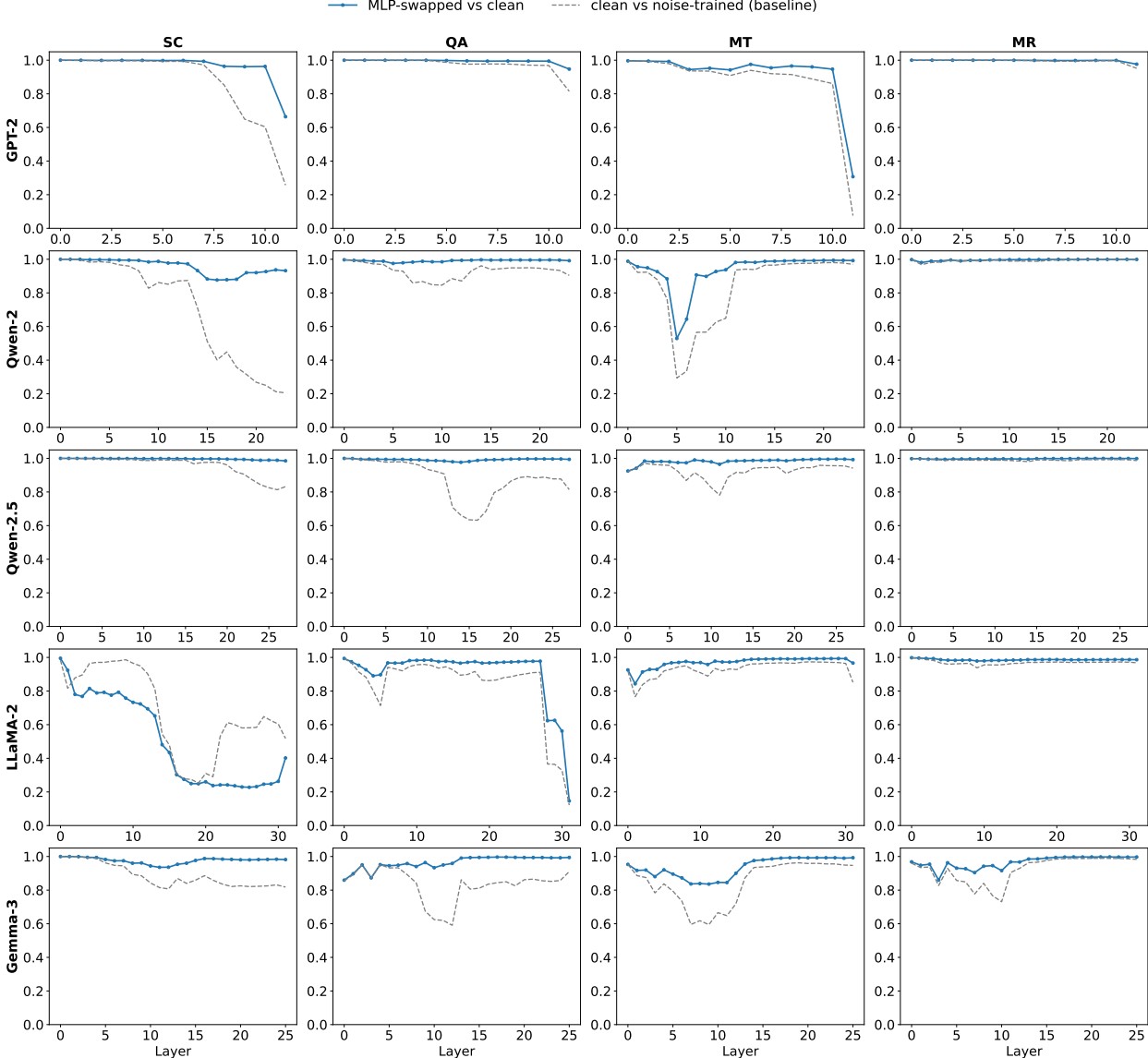

Figure 6: Layer-wise linear CKA after substituting all MLP sub-layers, shown for four tasks (columns: SC, QA, MT, MR) and five models (rows: GPT-2, Qwen2, Qwen-2.5, Llama-2, Gemma3). For each cell we replace every MLP sub-layer of the QLoRA noisy model under 40% LF noise with the corresponding sub-layer from the clean QLoRA-fine-tuned model, while keeping the attention and embedding parameters of the noise-trained model. The solid blue curve reports the per-layer linear CKA between this MLP-substituted model and the clean model; the gray dashed curve reports the linear CKA between the clean and noise-trained models (no substitution) and serves as the baseline. The x-axis indexes the layer at which CKA is measured. When the blue curve approaches 1 and remains well above the baseline, restoring the clean MLP weights is sufficient to recover the clean representation.

the lowest similarity point at each layer). **b)** As can be observed in Table 2 that LF has caused more damage compared to other noise, similarly the largest distortion (lowest layer CKA) is observed in LF noise for all the tasks across all the models in Figure 5. **c)** As it can be seen in Figure 4 that for most of the tasks the task-specific information was encoded in the later layers. Similar things can also be observed in Figure 5. In addition to the clean-vs-noise curve, Figure 5 reports two same-distribution baselines that the average linear CKA between independently random-seed trained clean models (clean variants) and between random-seed

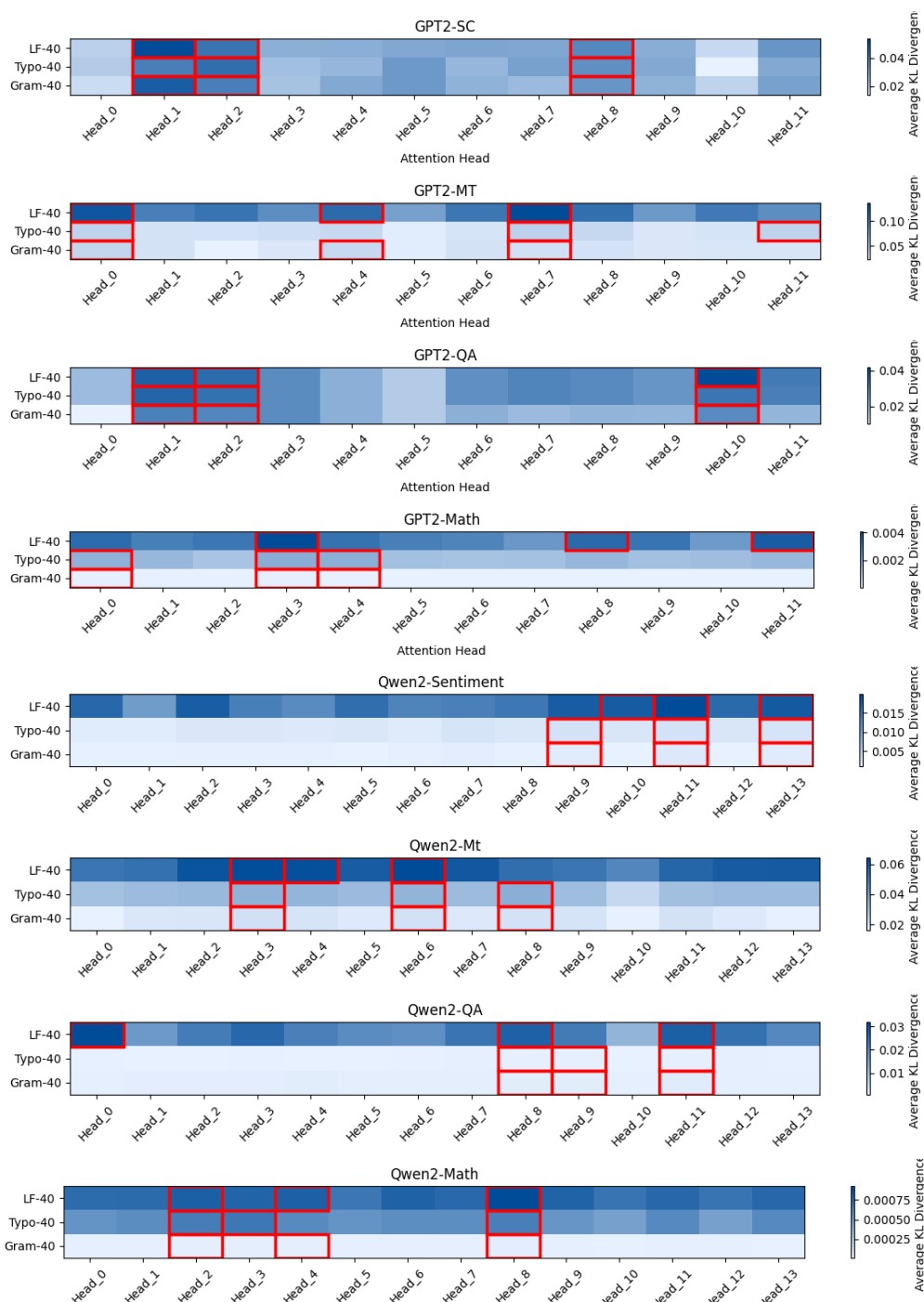

Figure 7: Attention Head Analysis for GPT-2 and Qwen 2-0.5B models in Sentiment CLassification (Sent), Machine Translation (MT) and Math Reasoning (Math).

noise-trained models (noisy variants). Both baselines remain high across all layers (layer-averaged CKA $\approx 0.99$ for clean variants and $\approx 0.98$ for noisy variants), whereas the clean-vs-noise curve falls clearly below them. The high clean-variant baseline shows that the observed drop is not an artefact of random seed variation, and the high noisy-variant baseline shows that the distortion is systematic and reproducible: two independently noise-trained models converge to nearly the same distorted representation rather than to

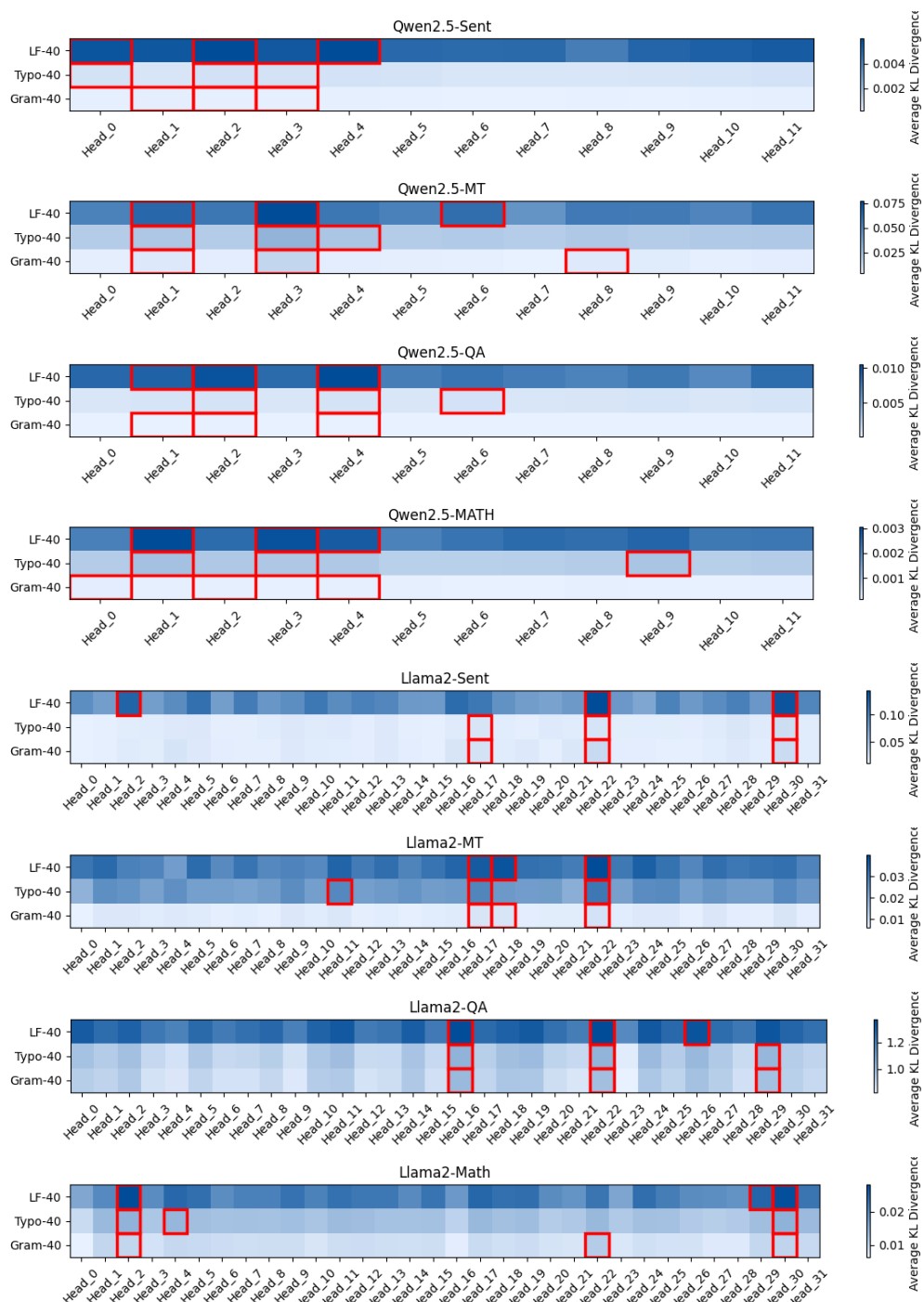

Figure 8: Attention head Analysis similar to Figure 7 for LLama2-7B and Qwen-2.5-1.5B Instruct model.

random ones. The CKA values of the initial layers are generally higher than the later layers except MT in Qwen2-0.5B. Based on the above-mentioned observation, it can be said that generally the noise affects the layers that have more task-specific information more compared to the ones where there was not that much task-specific information. The result for centered cosine similarity is shown in Figure 13 (Appendix F). The patterns are consistent with the CKA-based similarity results, confirming that noise primarily affects layers with more task-specific information. To address this, we added a new experiment in which we replaced

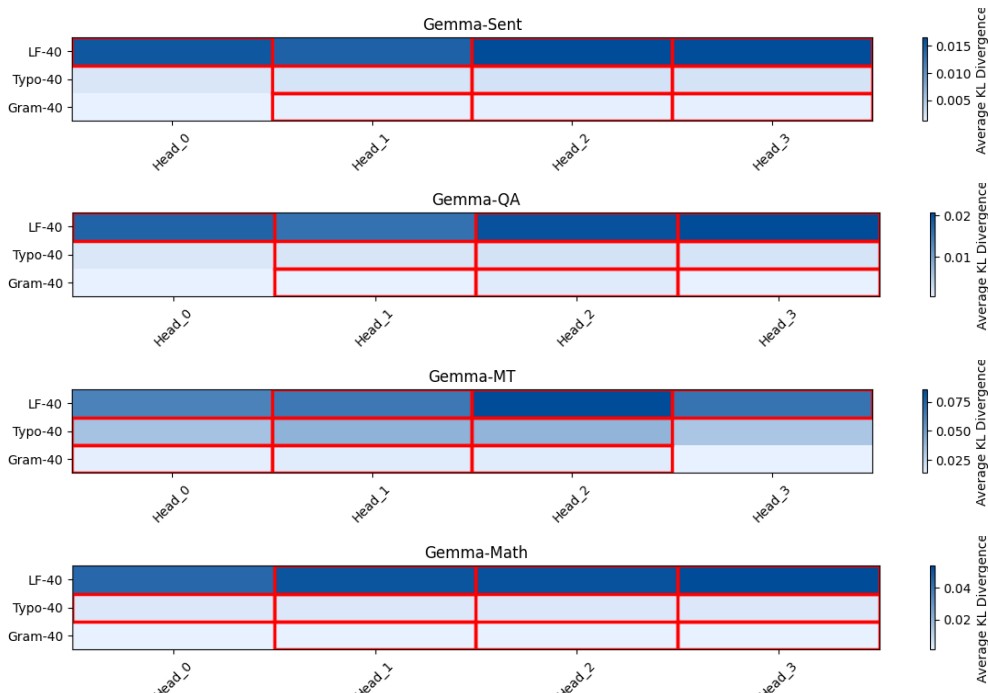

Figure 9: Attention head Analysis similar to Figure 7 for Gemma3-1B Instruction Tuned Model.

the MLP layers of the noisy model with those from the clean model and examined whether the change in layerwise representation persisted.

To further analyse whether feed-forward layers are primarily responsible for the damage in layerwise representations, we did an experiment where MLP layers of the noisy model was swapped with the clean model to isolate the effect of damage of all other components except MLP layer. It can be observed from Figure 6, when the MLP layers are replaced, the layerwise representations of the noisy model are more similar to the clean model compared the noisy model version where MLP layers were not swapped for Qwen2-0.5B, Qwen-2.5-1.5B Instruct and Gemma3-1B instruction tuned model. However, for GPT-2 and Llama2-7B the decrease in similarity has only increased from the swapped version to the noisy model version where no MLP layers were swapped. Based on the above observations, it can be concluded that whether MLP layers are primarily responsible for disrupting layer-wise representations is model-dependent.

## 6    Robust vs. Vulnerable Stratification

We initially investigated the overall changes in models through the methods described in Section 3.1, 3.2 and 3.3. However, the effect on noisy data fine-tuning may not be uniform on all the test samples. There are test samples for which the prediction of model remains unchanged after fine-tuning with noisy data. Broadly speaking, they are robust samples with respect to a task and a model. Similarly, there are samples for which predictions changed due to using a model fine-tuned on noisy data. Broadly speaking, these are vulnerable samples.

To examine the effect of robust and vulnerable samples separately, we stratified the evaluation samples into two groups: *robust* samples and *vulnerable* samples. We then applied all the above mentioned analysis approaches on the different types of dataset separately. The objective was to observe whether aggregate representational metrics mask heterogeneous effects across subpopulations with different type of outcomes.

From Figure 14, 16 in Appendix G we observed that in most cases the damage is more for the data points for which the prediction was wrong due to noise compared to the ones for which the prediction didn't change in

spite of noise. It is interesting to note that in spite of no change in prediction, there was still distortion in the internal representation.

## 7 Conclusion and Future Work

This work presents a systematic analysis of the effects of three types of noise (label noise, typographical noise, and grammatical noise) across four widely used NLP tasks and five different language models. Through a set of complementary analyses, we examine how these noise sources affect model behavior at both the prediction and representation levels. Our results show that the impact of noise tends to be largely localized within specific layers of the model rather than uniformly affecting the entire network. Furthermore, among the three noise types considered, label noise consistently leads to the most significant degradation in model performance, highlighting the sensitivity of LLMs to incorrect supervision signals during training. A further stratification of test samples into robust and vulnerable groups reveals that while vulnerable samples consistently show greater representational distortion, even robust samples whose predictions remain unchanged exhibit non-trivial internal representation shifts, indicating that task-level performance alone underestimates the true extent of noise-induced representational change.

These findings offer insights into how various forms of noise impact internal representations and task performance in LLMs. In future work, we plan to leverage these insights to design fine-tuning strategies that explicitly account for noise during training. In particular, we aim to develop robust fine-tuning approaches that can mitigate the adverse effects of noisy data while preserving task-relevant representations, thereby improving the reliability of LLMs in real-world noisy environments.

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

## A  Dataset Statistics and Prompt Templates

**Dataset statistics.**  Table 3 summarises the datasets and split sizes used for each task. All models share the same training, validation, and test samples for a given task. Concretely, we use the Yelp-Polarity *test* split for SC, a held-out slice of the Tatoeba EN–FR corpus disjoint from training for MT, and the GSM8K *test* split for MR. For QA we use SQuAD v1.1. Since its official test split is not publicly labelled, we draw all splits from the publicly labelled data: the training set and our validation set are taken from the SQuAD *train* split (the first 10,000 and the next 1,000 examples, respectively), while our held-out test set is the first 1,000 examples of the SQuAD *validation* split. The three subsets are disjoint.

Table 3: Dataset statistics for each task.

| Task | Source | Train | Val | Test |
|------|--------|-------|-----|------|
| Sentiment | Yelp Polarity | 10 000 | 1 000 | 1 000 |
| QA | SQuAD v1.1 | 10 000 | 1 000 | 1 000 |
| MT | Tatoeba EN–FR | 20 000 | 1 000 | 1 000 |
| MR | MetaMathQA / GSM8K | 20 000 | 1 000 | 1 000 |

**Prompt templates.**  All tasks are formatted as causal language modelling with a prompt–completion structure. Table 4 lists the prompt template used for each model–task combination. During training under the SFT-MASK protocol, the loss is computed only over the completion tokens (shown after the final colon). Prompt templates were selected per model to match each architecture's pre-training conventions. Crucially, the prompt is held constant across all noise conditions for a given model–task pair, ensuring that observed performance differences reflect only the effect of training data corruption.

Table 4: Prompt templates by model and task. The completion begins after the final colon in each template. `\n` denotes a newline character; long templates are wrapped at `\n\n` boundaries for readability. All Qwen-2.5-Instruct and Gemma-3-1B-IT templates share a common prefix `### Instruction:\n{instr}\n\n`, where `{instr}` is the task-specific instruction string: Sentiment: *"Classify the sentiment of the following sentence as Positive or Negative."*; QA: *"Answer the question based on the given context."*; MT: *"Translate the following English text to French."*; MR: *"Solve the math problem step by step."*. Templates shown for Qwen-2.5-Instruct and Gemma-3-1B-IT below omit this prefix.

| Task | Model | Template |
|------|-------|----------|
| Sentiment | GPT-2, Llama-2, Qwen-2 | `Review: {text}\nSentiment: {label}` |
| | Qwen-2.5-Instruct, Gemma-3-1B-IT | `### Input:\n{text}\n\n### Response: {label}` |
| QA | GPT-2 | `Context: {c}\nQuestion: {q}\nAnswer: {a}` |
| | Llama-2, Qwen-2 | `### Context:\n{c}\n\n`
`### Question:\n{q}\n\n### Answer: {a}` |
| | Qwen-2.5-Instruct, Gemma-3-1B-IT | `### Context:\n{c}\n\n`
`### Question:\n{q}\n\n### Response: {a}` |
| MT | GPT-2, Llama-2 | `English: {eng}\nFrench: {fra}` |
| | Qwen-2 | `Translate English to French.\n\n`
`### English:\n{eng}\n\n### French: {fra}` |
| | Qwen-2.5-Instruct, Gemma-3-1B-IT | `### Input:\n{eng}\n\n### Response: {fra}` |
| MR | GPT-2, Llama-2, Qwen-2 | `### Question:\n{q}\n\n### Solution: {solution}` |
| | Qwen-2.5-Instruct, Gemma-3-1B-IT | `### Input:\n{q}\n\n### Response: {solution}` |

## A.1 Noise Type Examples

Tables 5–7 show concrete before-and-after examples for each noise type applied to the three tasks. Corrupted portions are shown in **bold**.

Table 5: Label flip noise examples. The input text remains unchanged; only the target label/output is replaced.

| Task | Original | After Label Flip |
|---|---|---|
| Sentiment | *Text:* "The food was terrible and the service was even worse."
*Label:* Negative | *Text:* "The food was terrible and the service was even worse."
*Label:* **Positive** |
| QA | *Context:* "The Eiffel Tower was built in 1889 for the World's Fair. It is located in Paris, France."
*Question:* "When was the Eiffel Tower built?"
*Answer:* "1889" | *Context:* (unchanged)
*Question:* (unchanged)
*Answer:* **"the 10th century"** |
| MT | *English:* "Swimming at night is dangerous."
*French:* "Il est dangereux de nager de nuit." | *English:* (unchanged)
*French:* **"C'est la saison des fraises."** |
| MR | *Question:* "If Katerina buys 3 pots and 4 pans at the home goods store, with each pot costing \$20, and the total cost of her items is \$100, what is the cost of 2 pans assuming each pan is the same price?"
*Solution:* "3 pots cost \$60. 4 pans cost \$100 - \$60 = \$40, so each pan is \$10. Then 2 pans cost \$20. The answer is 20." | *Question:* (unchanged)
*Solution:* **"The cost of 3 pots is \$20 × 3 = \$60. The cost of 4 pans is \$100 - \$60 = \$40. Since each pan is the same price, the cost of 2 pans is \$40 / 4 = \$10. The answer is 10."** |

Table 6: Typo noise examples. Character-level perturbations (deletion, swap, insertion, or substitution) are applied to randomly selected words at a rate of 10% of words per sample.

| Task | Original | After Typo Injection |
|---|---|---|
| Sentiment | "The food was terrible and the service was even worse." | "The **fodo** was terrible and the service was even **wrse**." |
| QA | *Context:* "The Eiffel Tower was built in 1889 for the World's Fair." | *Context:* "The Eiffel **Towr** was built in 1889 for the World's Fair." |
| MT | *English:* "Swimming at night is dangerous." | *English:* "**Swimmiing** at night is dangerous." |
| MR | *Question:* "Tom has 12 marbles. He gives 5 to a friend. How many marbles are left?" | *Question:* "Tom has 12 **marlbes**. He gives 5 to a **frined**. How many marbles are **lft**?" |

Table 7: Grammatical noise examples. Rule-based substitutions target verb conjugation (*is↔are*, *was↔were*, *has↔have*) and article usage (*a↔an*) at a rate of 15% of words per sample.

| Task | Original | After Grammatical Errors |
|---|---|---|
| Sentiment | "The food was terrible and the service was even worse." | "The food **were** terrible and the service **were** even worse." |
| QA | *Context:* "The Eiffel Tower was built in 1889. It is located in Paris." | *Context:* "The Eiffel Tower **were** built in 1889. It **are** located in Paris." |
| MT | *English:* "Swimming at night is dangerous." | *English:* "Swimming at night **are** dangerous." |
| MR | *Question:* "Tom has 12 marbles. He gives 5 to a friend. How many marbles are left?" | *Question:* "Tom **have** 12 marbles. He gives 5 to **an** friend. How many marbles **is** left?" |

## B  Training Hyperparameters

All models are fine-tuned with the AdamW optimiser and a cosine learning-rate schedule and (4-bit NF4 quantisation) with LoRA adapters. Table 9 lists the LoRA configuration for each model–task pair. Table 10 lists the remaining training hyperparameters.

## C  Decoding Settings

All evaluations use **greedy decoding** (`do_sample` = False); we use no sampling, beam search, temperature, top-$k$, or top-$p$. For QA and MT we additionally apply a repetition penalty of 1.2 for Llama-2 and Qwen2 to suppress degenerate repetition; GPT-2 and Qwen-2.5-Instruct use no repetition penalty. The per-task generation budget (`max_new_tokens`) and repetition penalty are summarised in Table 8.

Table 8: Decoding settings used at evaluation. All tasks use greedy decoding (`do_sample` = False).

| Task | max_new_tokens | Repetition penalty |
|---|---|---|
| SC (Sentiment) | 5 | none (all models) |
| QA | 50 | 1.2 for Llama-2 & Qwen; none for GPT-2, Qwen-2.5 & Gemma-3 |
| MT | 50 (GPT-2: 60) | 1.2 for Llama-2 & Qwen; none for GPT-2, Qwen-2.5 & Gemma-3 |
| MR (Math) | 512 | none (all models) |

Table 9: LoRA adapter configuration. "All 7" denotes `q_proj`, `k_proj`, `v_proj`, `o_proj`, `gate_proj`, `up_proj`, `down_proj`.

| Model | Task | $r$ | $\alpha$ | Dropout | Target Modules |
|---|---|---|---|---|---|
| GPT-2 | All | 16 | 32 | 0.05 | c_attn, c_proj, c_fc |
| Qwen2 | Sentiment | 8 | 32 | 0.10 | All 7 |
| | QA | 16 | 32 | 0.05 | All 7 |
| | MT | 32 | 64 | 0.05 | All 7 |
| | MR | 16 | 32 | 0.05 | All 7 |
| Qwen-2.5-Instruct | Sentiment | 8 | 32 | 0.10 | All 7 |
| | QA | 16 | 32 | 0.05 | All 7 |
| | MT | 32 | 64 | 0.05 | All 7 |
| | MR | 16 | 32 | 0.05 | All 7 |
| Llama-2 | Sentiment | 32 | 64 | 0.05 | All 7 |
| | QA | 16 | 32 | 0.05 | All 7 |
| | MT | 128 | 256 | 0.05 | All 7 |
| | MR | 16 | 32 | 0.05 | All 7 |
| Gemma-3-1B-IT | Sentiment | 8 | 32 | 0.10 | All 7 |
| | QA | 16 | 32 | 0.05 | All 7 |
| | MT | 32 | 64 | 0.05 | All 7 |
| | MR | 16 | 32 | 0.05 | All 7 |

## D  QLoRA vs Full Fine-Tuning Attention Shift Comparison

Figure 10 compares the same KL extraction under two fine-tuning regimes: QLoRA (4-bit base + LoRA adapter on attention and MLP projections, $r = 16$) and full FT (all parameters updated, learning rate $2 \times 10^{-5}$).

The relative magnitudes of QLoRA-driven and FFT-driven attention drift are *not* consistent across models: GPT-2 generally shows larger QLoRA drift (1.9×–6.1× over FFT across SC/QA/MT/Math), whereas Qwen2 shows the opposite trend on three of four tasks (FFT drift is up to 13× larger). Llama-2 QA is the most

Table 10: Training hyperparameters. All configurations use the AdamW optimiser and a cosine LR schedule. Effective batch = per-device batch × gradient-accumulation (single GPU). Weight decay is 0 except where noted.

| Model | Task | LR | Eff. Batch | Epochs | Warmup | Max Len |
|---|---|---|---|---|---|---|
| GPT-2 | Sentiment | 1e-4 | 32 | 3 | 0.03 | 256 |
| | QA | 5e-5 | 32 | 3 | 0.03 | 1024 |
| | MT | 1e-4 | 32 | 30 | 0.03 | 256 |
| | MR | 5e-5 | 32 | 3 | 0.03 | 1024 |
| Qwen2 | Sentiment | 5e-5 | 80 | 3 | 0.03 | 256 |
| | QA | 2e-5 | 32 | 2 | 0.03 | 1024 |
| | MT | 2e-4 | 32 | 3 | 0.03 | 256 |
| | MR | 2e-5 | 32 | 2 | 0.03 | 1024 |
| Qwen-2.5-Instruct | Sentiment | 5e-5 | 64 | 3 | 0.03 | 256 |
| | QA | 2e-5 | 32 | 2 | 0.03 | 1024 |
| | MT | 2e-4 | 32 | 3 | 0.03 | 256 |
| | MR | 2e-5 | 32 | 2 | 0.03 | 1024 |
| Llama-2 | Sentiment | 1e-4 | 16 | 3 | 0.05 | 256 |
| | QA | 2e-4 | 8 | 2 | 0.03 | 1024 |
| | MT | 1e-4 | 32 | 10 | 0.05 | 128 |
| | MR | 2e-4 | 8 | 2 | 0.03 | 1024 |
| Gemma-3-1B-IT | Sentiment | 5e-5 | 64 | 3 | 0.03 | 256 |
| | QA | 2e-5 | 32 | 2 | 0.03 | 1024 |
| | MT | 2e-4 | 32 | 3 | 0.03 | 256 |
| | MR | 1e-4 | 32 | 2 | 0.03 | 1024 |

striking single cell: QLoRA mean KL is 1.27 versus only 0.29 under FFT, which is a 4.4× larger reaction to the same noisy labels. We interpret this as an architecture-specific interaction: the low-rank adapter, when confronted with hard supervision noise, appears to adapt attention patterns more aggressively than a fully updated network whose larger MLP capacity can absorb the noise without restructuring attention.

## E  Attention Pattern Comparison with Seed variation as Baseline

Figure 11 compares attention-pattern changes between the clean and noisy fine-tuned models, as well as across different seed variants of the clean fine-tuned model. The seed-variant clean models exhibit only minor changes in attention values, mostly on the order of 0.05. Therefore, the stronger attention-pattern changes observed in the clean-vs-noisy comparison are primarily attributable to the injected noise. Similar pattern is observed for Figure 12 where the difference in the seed variations of noisy (with 40% label flip noise) model is used as baseline.

## F  Layer-wise Centered Cosine Similarity

Figure 13 presents the full layer-wise centered cosine similarity between clean and noise-trained models across all nine model–task combinations. Centered cosine similarity is computed after subtracting the sample-wise mean from each representation matrix, removing the anisotropy-induced bias that inflates raw cosine similarity Ethayarajh (2019).

Table 11: Supervised full fine-tuning task-related performance of different models under 40% noise ratio (SC: Accuracy; QA: F1 score; MT: BLEU score, MR: Exact Match).

| Task | Model | Baselines | | Label Flip 40% | Typo Error 40% | Gramm. Error 40% |
|---|---|---|---|---|---|---|
| | | Pretrained | Clean FT | | | |
| Full fine-tuning | | | | | | |
| Sentiment | GPT-2 | 0.12 | 91.33 | 75.51 | 90.91 | 91.41 |
| | Qwen2 | 16.30 | 94.30 | 51.80 | 94.60 | 94.60 |
| | Llama-2 | 0.60 | 93.50 | 96.40 | 93.90 | 94.30 |
| | Qwen2.5-1.5B-Instruct | 92.70 | 95.00 | 92.50 | 94.70 | 95.10 |
| QA | GPT-2 | 8.11 | 35.51 | 31.58 | 43.97 | 35.72 |
| | Qwen2 | 38.75 | 76.47 | 68.24 | 75.47 | 76.85 |
| | Llama-2 | 30.90 | 83.71 | 46.67 | 83.20 | 83.4 |
| | Qwen2.5-1.5B-Instruct | 53.8 | 87.2 | 81.6 | 87.03 | 87.11 |
| MT | GPT-2 | 0.31 | 5.08 | 1.84 | 4.79 | 5.00 |
| | Qwen2 | 14.49 | 43.93 | 41.18 | 43.87 | 43.72 |
| | Llama-2 | 20.67 | 50.24 | 48.83 | 50.11 | 50.00 |
| | Qwen2.5-1.5B-Instruct | 18.87 | 45.5 | 33.7 | 45.13 | 46.23 |
| Math Reasoning | GPT-2 | 1.2 | 2 | 2 | 2.3 | 2.1 |
| | Qwen2 | 10.1 | 41 | 39.5 | 39.7 | 41.8 |
| | Llama-2 | 5.2 | 54.1 | 50.9 | 54.3 | 50.9 |
| | Qwen2.5-1.5B-Instruct | 19.6 | 70.1 | 68.9 | 69.1 | 70 |

## G  Robust Vs. Vulnerable Stratification

This appendix presents the full stratification results for robust and vulnerable samples across all model–task combinations under label-flip noise. Robust samples are those whose predictions remain unchanged after fine-tuning with noisy data, while vulnerable samples are those whose predictions change. Figure 14 shows the centered cosine similarity by group . In most cases, vulnerable samples show lower centered cosine similarity than robust samples at deeper layers, indicating greater representational distortion for samples whose predictions are affected by noise. However, it is interesting to note that even robust samples show non-trivial representational distortion despite their predictions remaining unchanged. Figure 16 shows the Linear CKA by group. The vulnerable–robust gap is most pronounced for Qwen2 sentiment at 40% noise, where vulnerable CKA drops to 0.260 compared to 0.612 for robust samples at the final layer. An exception is observed for Llama-2 QA, where vulnerable CKA slightly exceeds robust CKA across multiple layers and noise levels, reversing the expected direction. This anomaly may reflect the optimization instability of Llama-2 near the critical noise threshold, where small changes in training conditions can lead to very different outcomes.

## H  Clean-vs-Clean CKA Baseline

To establish a ceiling for CKA and confirm that noise-induced CKA drops are not attributable to seed variance alone, we compute CKA between clean models trained with different random seeds.

The clean-vs-clean CKA floor of 0.890 is far above the noise-condition values (0.11–0.42), confirming that the representational changes reported in our main experiments reflect genuine noise effects rather than stochastic training variance.

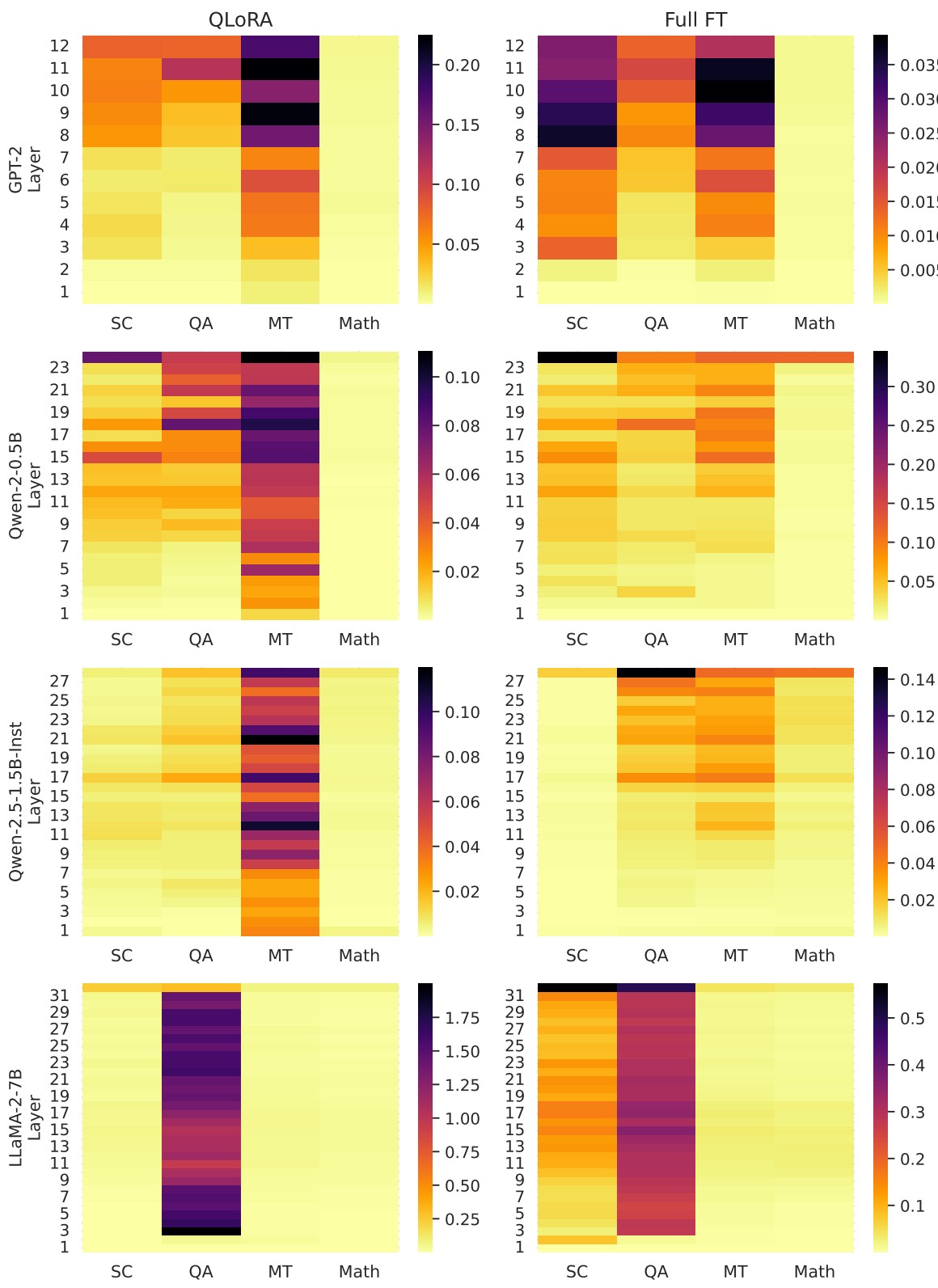

Figure 10: Attention KL under LF40 for each (model, task) pair under QLoRA (left column of each row) and full FT (right column). Heatmap color is the mean KL divergence across heads; same color scale within each panel.

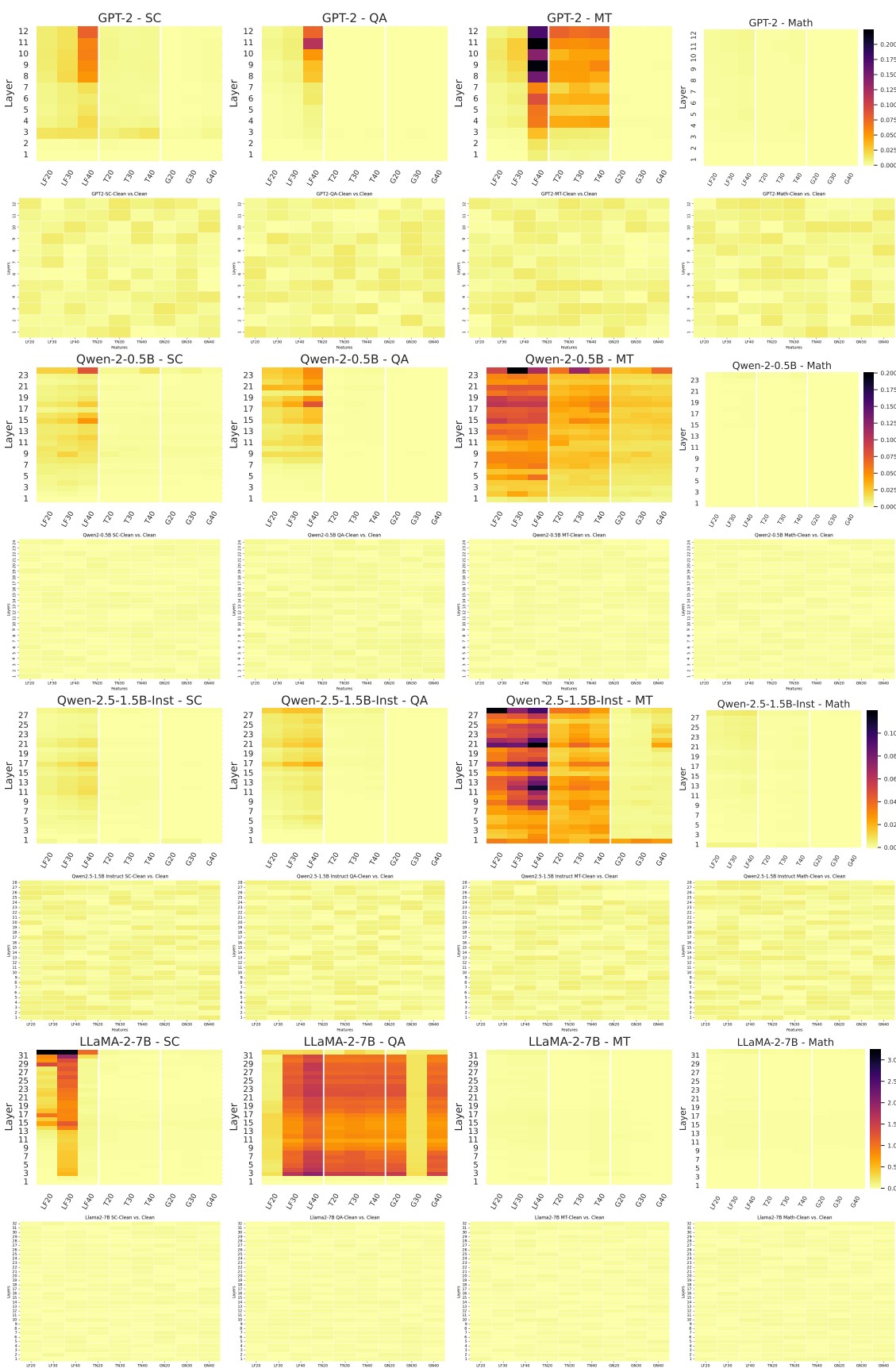

Figure 11: Comparison of attention-pattern changes across clean fine-tuned, noisy fine-tuned, and alternative attention models under different seed variations. Random seed variations are shown in every alternative row with 'clean vs.clean' title.

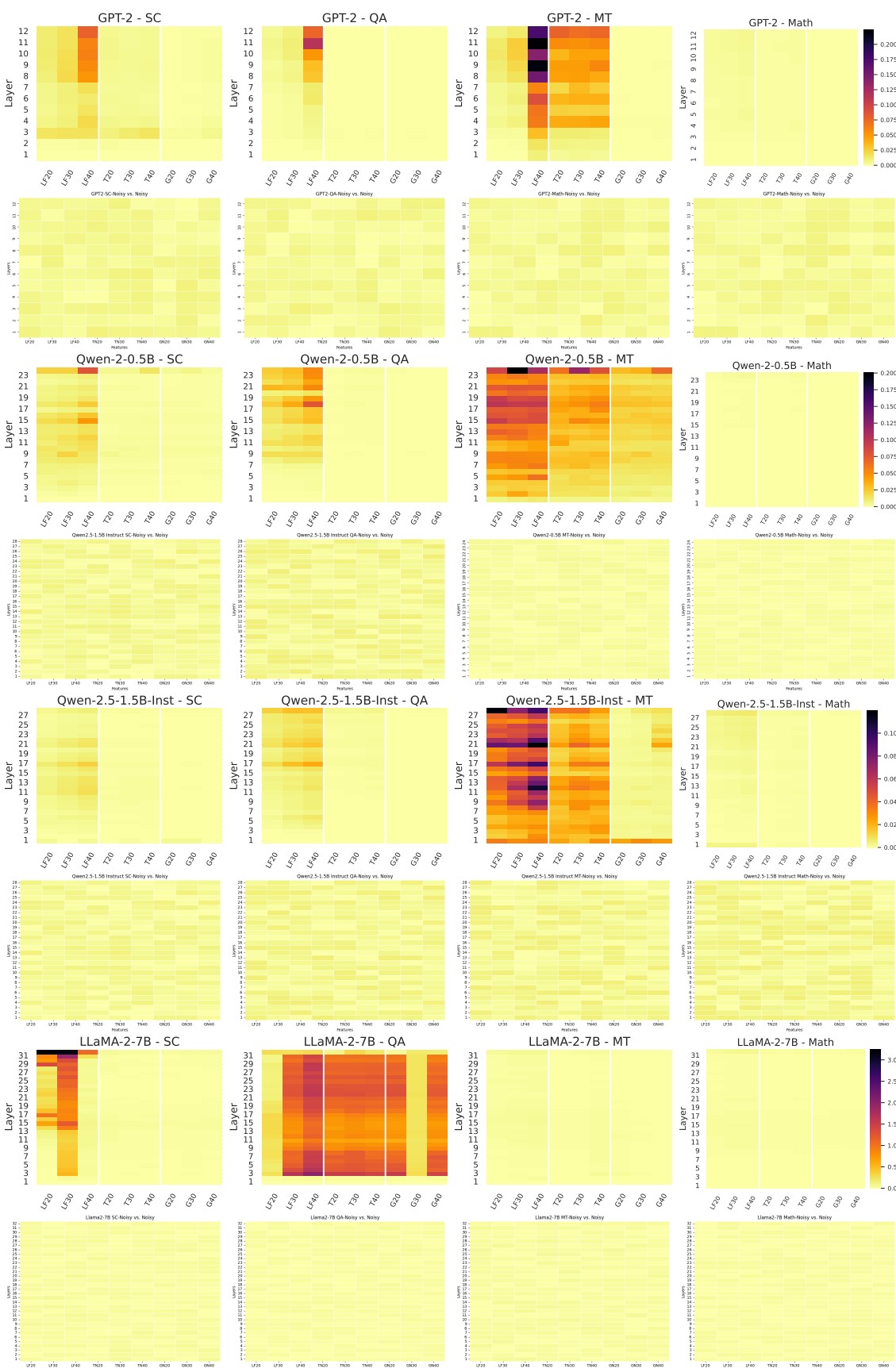

Figure 12: Comparison of attention-pattern changes across clean fine-tuned, noisy fine-tuned, and alternative attention models under different seed variations. Random seed variations are shown in every alternative row with 'noisy vs. noisy' title.

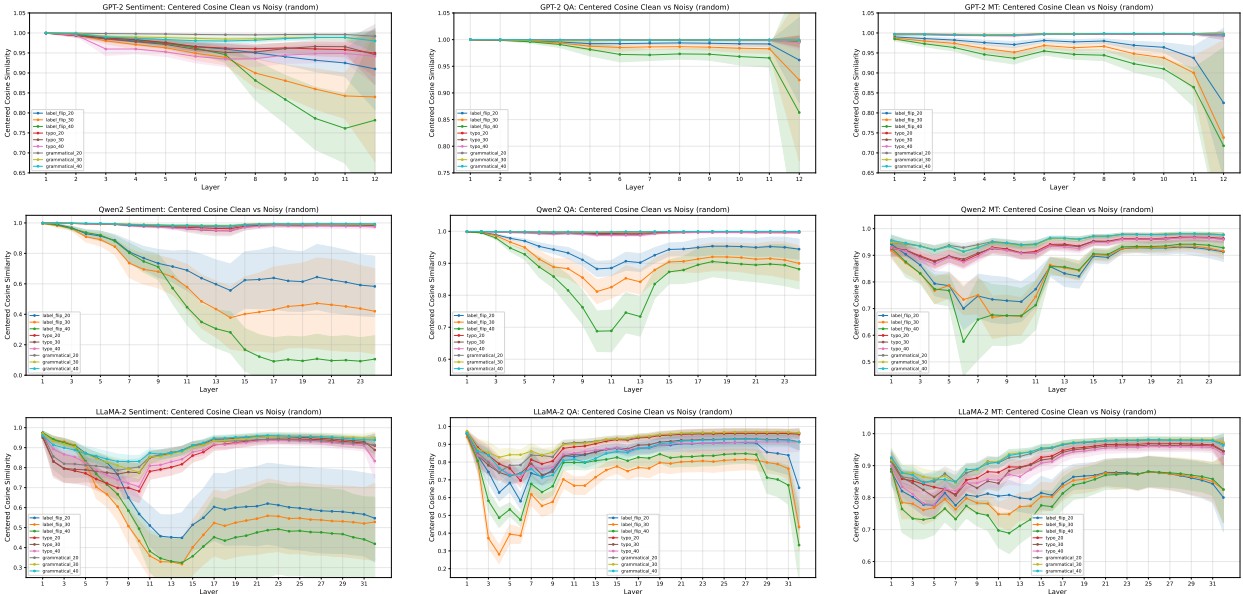

Figure 13: Layer-wise centered cosine similarity between clean and noise-trained model representations across all nine model–task combinations. Rows correspond to models (GPT-2, Qwen2, Llama-2); columns correspond to tasks (SC, QA, MT). Centered cosine removes the shared mean direction before computing similarity, correcting for the anisotropy of contextualised representations.

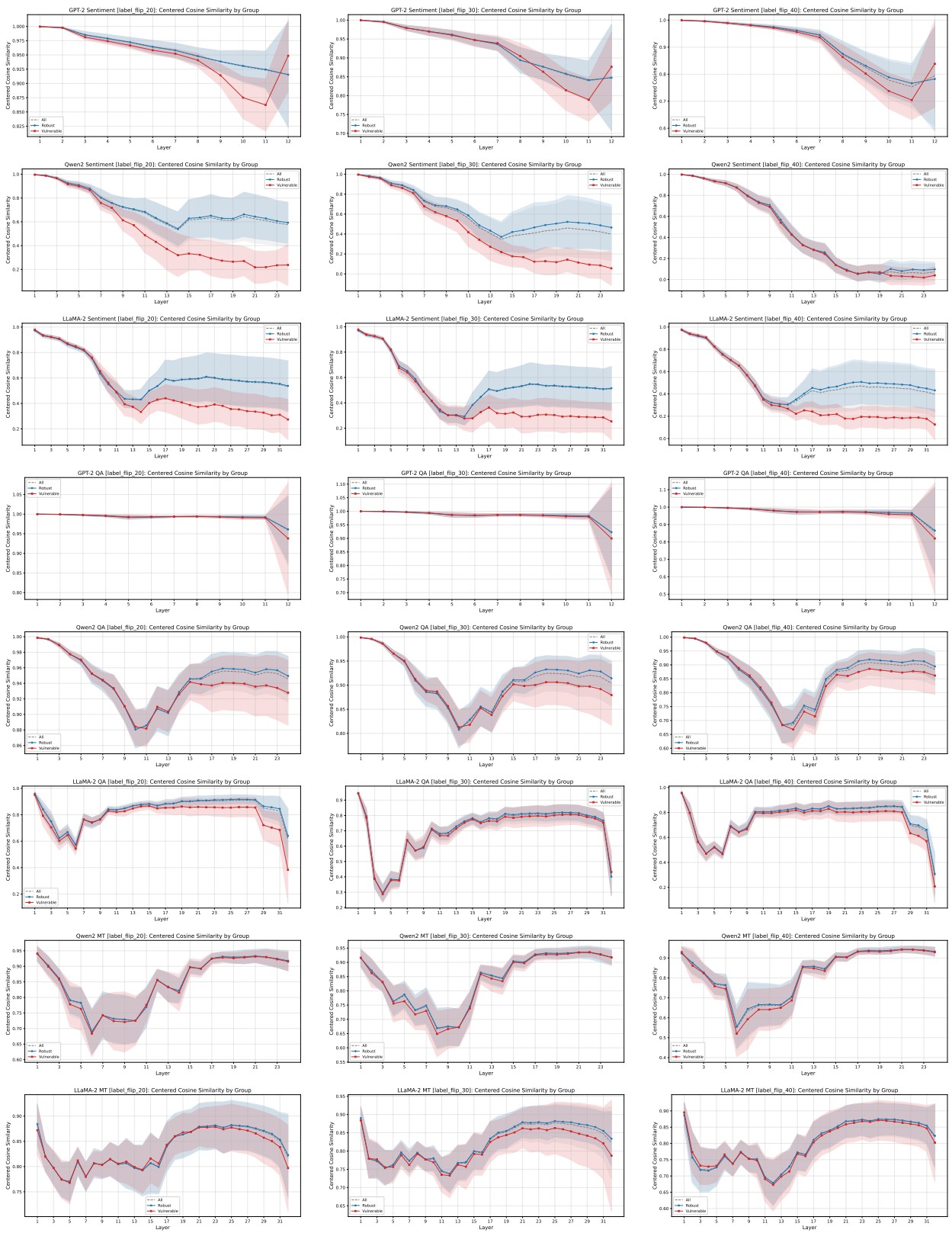

Figure 14: Robust vs. vulnerable stratification: **centered cosine similarity** for **SC**, **QA** and **MT** under label-flip noise. Centered cosine removes the shared mean direction before computing similarity, correcting for anisotropy.

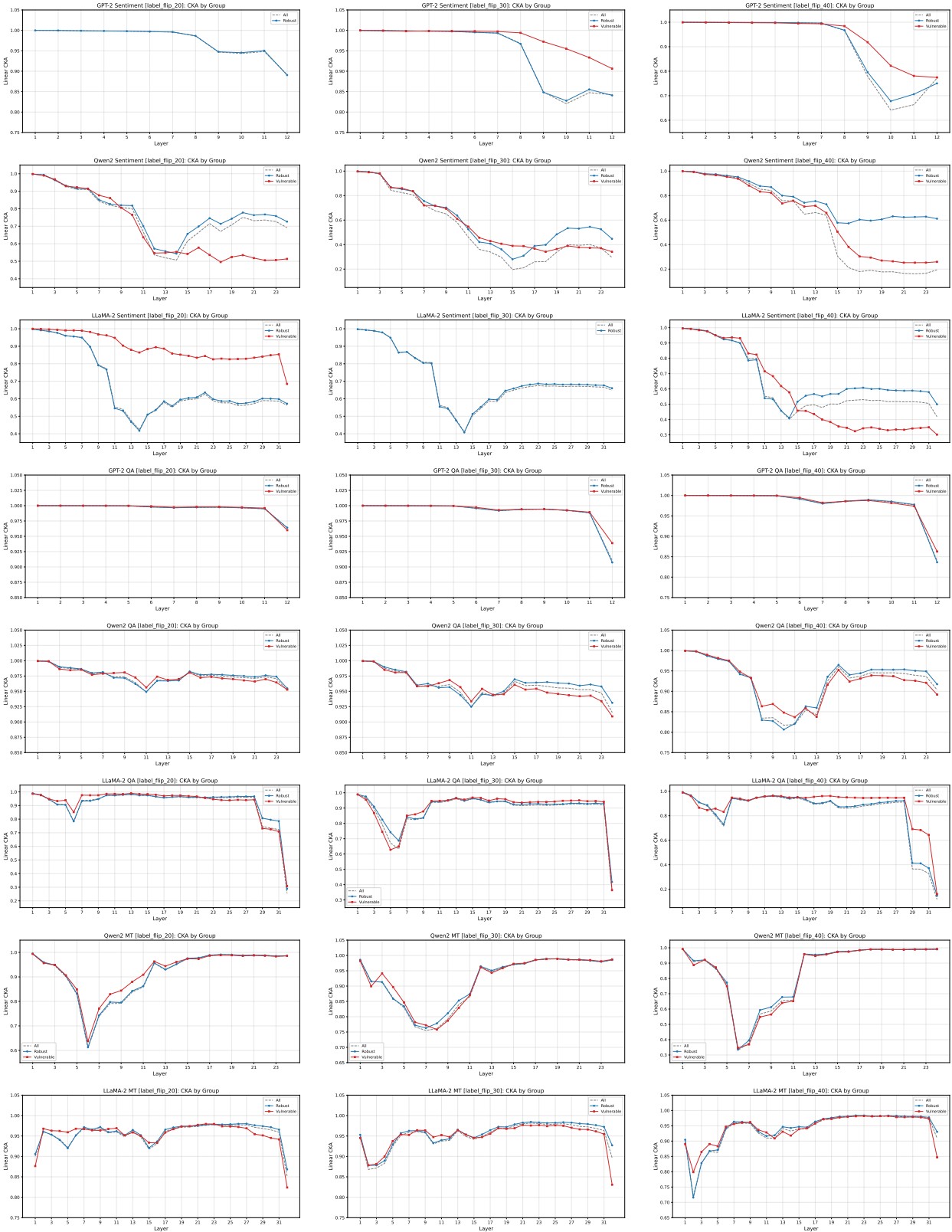

Figure 15: Robust vs. vulnerable stratification: **Linear CKA** for **SC**, **QA** and **MT** under label-flip noise. CKA captures inter-sample relational structure.

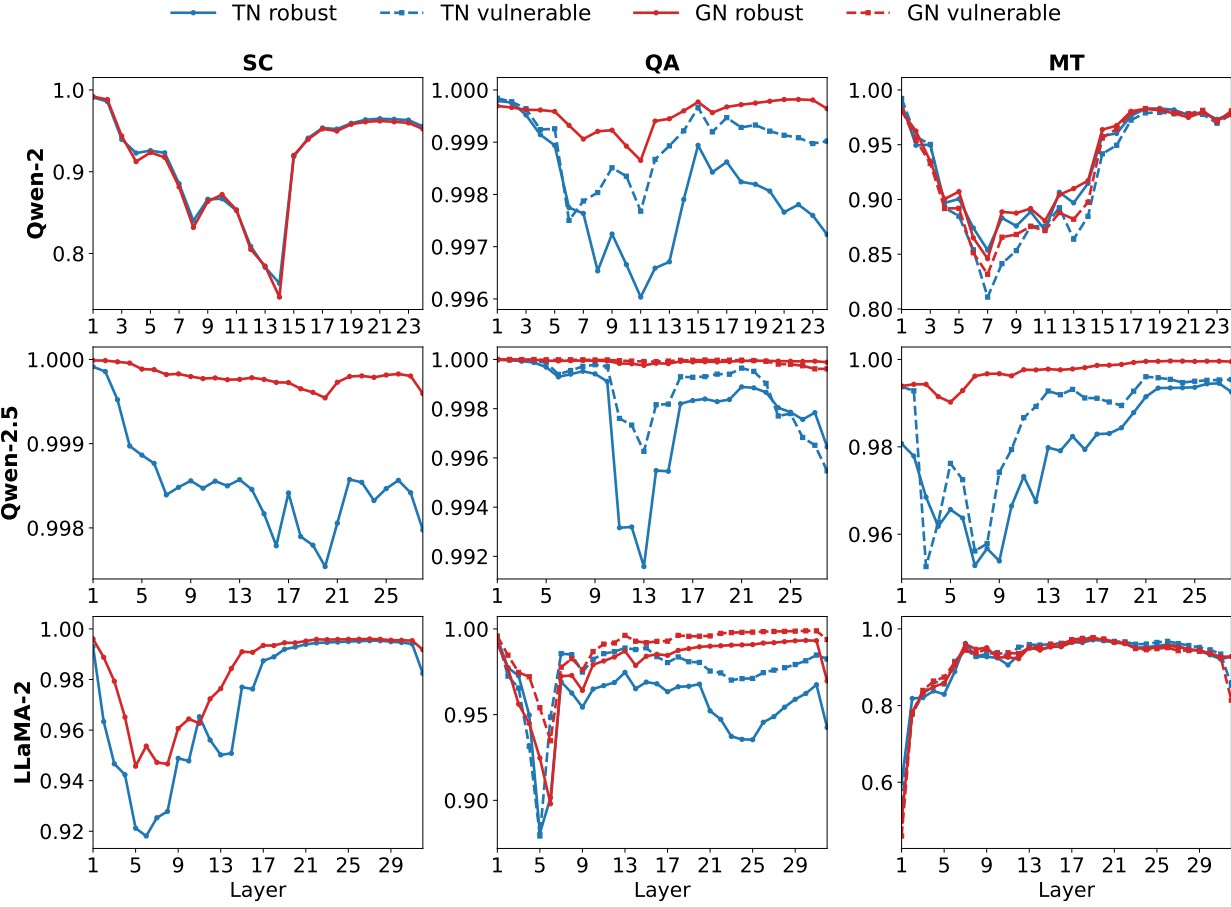

Figure 16: Robust vs. vulnerable stratification: **Linear CKA** for **SC**, **QA** and **MT** under typo noise and grammatical noise. CKA captures inter-sample relational structure. For each model–task–noise variant, we use the most-damaging corruption ratio.

