# OpenReview forum: "Analyzing the Effect of Noise in LLM Fine-tuning"
_TMLR — Under review for TMLR_

### Review · Reviewer_A2pz · 2026-05-09

**Summary Of Contributions:**

This paper studies how different types of noise affect LLM fine-tuning across three NLP tasks: sentiment classification, question answering, and machine translation. The authors consider three model families, GPT-2, Qwen2, and Llama-2, and three types of noise: label flip noise, typographical noise, and grammatical noise. Beyond reporting task-level performance, the paper analyzes attention patterns through KL divergence and Spearman rank correlation, layer-wise task information through probing and Logit Lens metrics, and representation changes through cosine similarity and CKA.

**Strengths**:
- The paper addresses a practically relevant problem: fine-tuning data can contain different kinds of noise, and understanding how such noise affects LLM behavior is useful.
- The experimental scope is relatively broad. The authors consider three tasks, three model families, and three types of noise.
- The paper attempts to go beyond task-level performance by analyzing attention patterns, probing/logit-lens behavior, and representation similarity.

**Weaknesses**:
- Several main claims are stronger than what the current evidence supports. In particular, the conclusions about label noise being generally most harmful, attention patterns remaining stable, and noise effects being localized to task-specific layers require stronger justification and controls.
- Some experimental comparisons are confounded, such as target-side label/output corruption versus input-side typographical/grammatical perturbations, and full fine-tuning for GPT-2 versus QLoRA for the other models.
- Some analysis choices are under-justified, including position-wise attention comparisons for input-side perturbations and first-target-token MRR as a proxy for task information in QA/MT.
- The presentation needs improvement. Several central figures are difficult to read, and some citations appear as raw BibTeX keys in the prose.

**Audience:**

Yes

**Audience Explanation:**

Yes. The topic is potentially interesting to the TMLR audience. Noisy fine-tuning data is common in practice, and understanding how different types of noise affect LLM behavior is relevant for both robustness and model analysis.

**Broader Impact Concerns:**

I do not see major broader impact concerns.

**Claims And Evidence:**

No

**Claims Explanation:**

1. I am not convinced by the evidence for the claim that attention patterns remain stable. The attention analysis compares clean and noisy attention distributions using KL divergence and Spearman rank correlation. This is more interpretable for label-flip noise, where the input tokens are unchanged. However, for typographical and grammatical noise, the perturbations may change tokenization or token positions. Without an explicit token-alignment procedure, position-wise attention comparisons may reflect tokenization or positional mismatch rather than meaningful changes in attention behavior. In addition, the KL values are not calibrated against a baseline such as clean-vs-clean runs with different seeds, and high Spearman correlation only shows rank similarity, not necessarily stability of the full attention distribution.

2. The probing analysis for QA and MT relies primarily on Logit Lens MRR of the first target token. This is a weak proxy for task-relevant information in generative tasks. QA and MT outputs are sequence-level objects, and the first target token may be a high-frequency token, a subword fragment, or strongly influenced by the prompt/template. Although the appendix reports results over the first five target tokens, this only partially addresses the issue, especially for machine translation. The current evidence supports the narrower claim that some layers rank early reference tokens higher, not that they encode the full task solution.

3. The representation similarity analyses lack sufficient baselines. Independently fine-tuned models can differ in their representations even without noise. The paper needs stronger clean-vs-clean and noisy-vs-noisy comparisons across random seeds to establish what level of CKA or cosine change is meaningful. Otherwise, the observation that clean and noisy models have different hidden states is difficult to interpret. I also find the mechanistic interpretation that stable attention plus changed representations implies changes in feed-forward representations to be under-supported, since hidden-state changes can arise from many components of the transformer.

4. Cross-model comparisons are confounded by the use of different fine-tuning methods: GPT-2 is fully fine-tuned, while Qwen2 and Llama-2 use QLoRA. The provided GPT-2 LoRA ablation is too limited to resolve this concern, since it covers only one model, one task, and one noise condition.

**Requested Changes:**

Please see my comments above for the main concerns. In addition to the methodological issues discussed there, I would consider the following two changes important for improving the submission:

1. The authors should improve the readability of Figures 2, 3, and 4. These figures are central to the paper’s evidence, but they are currently difficult to read. The axis labels are too small, and too many conditions are overlaid or compressed into the same figure. Figure 4 is also heavily overplotted, with many noise conditions shown in each subplot. The authors should consider separating noise types into different panels, plotting deviations from the clean model, increasing font sizes, simplifying legends, or providing quantitative summaries.

2. The citation formatting in Section 2.1 should be corrected. Several references appear as raw BibTeX keys in the prose, e.g., “liu2020early,” “tanzer2022memorisation,” This is not standard citation formatting and should be fixed throughout the manuscript.

---

> ### Author Response · Authors · 2026-06-17
>
> Thank you for you detailed comments. We have considered all your suggestions and submitted a revised draft. Our response to your concerns is as follows.
>
> **Change in Token Position:** We agree that the number of tokens in a clean text and its corresponding corrupted version, with added typographical or grammatical noise, may differ. However, when comparing the attention matrices, we provide the exact same clean input to both the clean model and the noisy model. The noisy model was fine-tuned on noisy data, but it was tested on clean data. The objective of these experiments is to investigate how the clean model and the noisy model behave when given the same set of inputs. To clarify this confusion, we have also added the above clarifications in Section 3.1 and in Evaluation, Section 4 in the updated paper.
>
> **Probing Analysis for QA and MT** We have now updated the probing analysis for QA and MT. To capture the task level effectiveness, for the QA and MT tasks, we used a teacher-forced setting to generate the complete output string. Specifically, if the ground-truth output for an input contains (k) tokens, we apply teacher forcing (k) times to generate all (k) tokens. We then used F1 score for question answering and BERTScore for measuring the effectiveness of the machine translation models. We have updated the probing details in Section 3.2 in the updated draft.
>
> **Lack of Sufficient baseline for Representational Similarity**  We have now added clean-vs-clean and noisy-vs-noisy baselines, based on different random seeds, to analyse the CKA similarity between representations. As shown in Figure 5, the similarities between these variations across different seeds are very high (almost equals to 1 in most of the cases), with minimal fluctuation. Therefore, our previous conclusion still holds.
>
> **Mechanistic Interpretation of  Feedforward Layers Driving Representational Changes**  We agree that, in the previous version of the paper, there was insufficient evidence to support the claim that feed-forward representations were primarily responsible for the change in layerwise representation in the noisy model. To address this, we added a new experiment in which we replaced the MLP layers of the noisy model with those from the clean model and examined whether the change in layerwise representation persisted. It can be observed from Figure 6, when the MLP layers are replaced, the layerwise representations of the noisy model are more similar to the clean model  compared the noisy mdoel version where MLP layers were not swapped for Qwen2-0.5B and Qwen2.5-1.5B instruct model. However, for GPT-2 and Llama2-7B the decrease in similarity has only increased from the swapped version to the noisy model version where no MLP layers were swapped. Based on the above observations, it can be concluded that whether MLP layers are primarily responsible for disrupting layer-wise representations is model-dependent. We have updated our observation in Section 5.4 and we also removed the previous claim from conclusion.
>
> **Confounded Cross-Model Comparison** To address the issue of confounded comparisons, we have now added two sets of experiments. The first set reports QLoRA-based results, where all models are fine-tuned using QLoRA (Table 2, Figure 2). The corresponding full fine-tuning results are in Table 11 and Figure 8. The comparison of both sets of tables (performance on noisy data) and Figures (attention pattern analysis) shows a similar pattern.
>
> Please let us know if you have any further concerns.

---

> > ### Author Response · Authors · 2026-06-17
> >
> > **Clean vs. Clean or Noisy vs. Noisy Baseline for Attention pattern Analysis**  We have now added Figure 10 in APpendix which compares attention-pattern changes between the clean and noisy fine-tuned models, as well as across different seed variants of the clean fine-tuned model. The seed-variant clean models exhibit only minor changes in attention values, mostly on the order of 0.05. Therefore, the stronger attention-pattern changes observed in the clean-vs-noisy comparison are primarily attributable to the injected noise. Similar pattern is observed for Figure 11 where the difference in the seed variations of noisy (with 40\% label flip noise) model is used as baseline.

---

### Review · Reviewer_pg52 · 2026-05-15

**Summary Of Contributions:**

**Summary:**

This paper studies how different types of training noise affect LLM fine-tuning at both the performance and representation levels. The authors consider three noise types—label noise, typographical noise, and grammatical noise—across three tasks, namely sentiment classification, question answering, and machine translation, using GPT-2, Qwen2, and Llama-2. Beyond task-level metrics, the paper analyzes attention divergence, attention ranking stability, layer-wise probing or Logit Lens behavior, and representation similarity using cosine similarity and CKA. The main findings are that label noise causes the largest performance degradation, the impact of noise is mainly localized in task-relevant layers, and attention patterns remain relatively stable compared with hidden representations.



**Strengths:**

- **A meaningful and practically relevant problem:** Noise in fine-tuning data is common in real-world LLM adaptation pipelines, and understanding its effect is important for building reliable models. The paper studies this issue across multiple noise types, tasks, and model families, which makes the empirical setting reasonably broad. The main findings, especially that label noise is more harmful than input-side typographical or grammatical noise, are intuitive and mostly consistent across the reported experiments.

- **The analysis is relatively detailed and goes beyond task-level performance:** Instead of only reporting accuracy, F1, or BLEU, the authors analyze attention divergence, attention ranking stability, layer-wise probing or Logit Lens results, and representation similarity through cosine similarity and CKA. This is a useful aspect of the paper, as it attempts to connect changes in downstream performance with changes in the model’s internal layer-wise representations. These analyses provide more insight than a purely benchmark-driven study.



**Weaknesses:**

- **The choice of models is somewhat limited and dated:** The experiments are conducted on GPT-2, Qwen2-0.5B, and Llama-2-7B. While these models cover different scales and families, they may not fully reflect the behavior of more recent open-source LLMs. Including newer small or medium-sized instruction-capable models would make the conclusions more convincing and more relevant to current LLM fine-tuning practice.

- **The task setting is relatively simple and traditional:** The evaluated tasks, including sentiment classification, extractive question answering, and machine translation, are classic NLP benchmarks. However, they do not necessarily represent modern LLM fine-tuning scenarios such as instruction tuning, preference tuning, multi-turn dialogue, or tool-use tasks. As a result, the experiments are closer to supervised fine-tuning on traditional NLP tasks, and it is unclear how well the findings generalize to more realistic contemporary LLM adaptation settings.

- **The main conclusions are well supported but not surprising:** The observation that noise mainly affects middle-to-later layers is consistent with the common understanding that lower layers tend to capture more local lexical or syntactic patterns, while higher layers encode more contextual and task-specific semantic information. Similarly, the finding that attention patterns remain relatively stable is also intuitive. Many sink-like tokens are non-semantic structural tokens that mainly reflect a structural bias of the model, they can absorb redundant attention mass without necessarily contributing meaningful semantic information through the residual stream. Such attention patterns are therefore unlikely to change substantially under fine-tuning noise. As a result, although the empirical evidence is systematic, the conclusions do not substantially go beyond existing intuitions about layer-wise representation and at

**Audience:**

Yes

**Audience Explanation:**

Some TMLR readers would likely be interested in the findings, since noisy fine-tuning data is a common practical issue in LLM adaptation. Although the task settings are somewhat traditional and do not fully reflect the latest LLM fine-tuning scenarios, they are still valid benchmarks for studying the effect of different noise types. The systematic analysis of performance and internal representations may be useful for researchers working on robust fine-tuning, representation analysis, and reliable LLM training.

**Claims And Evidence:**

Yes

**Claims Explanation:**

The main claims are supported by clear and systematic empirical evidence across multiple models, tasks, noise types, and analysis metrics. Although the conclusions are mostly intuitive and do not substantially go beyond existing expectations, the experiments are well organized and provide consistent evidence for the reported findings, especially regarding the stronger impact of label noise and the relative stability of attention patterns.

**Requested Changes:**

In my view, the main limitation that prevents the paper from fully meeting the TMLR standard is that the experimental setting is not sufficiently representative of current LLM fine-tuning practice. The current study is systematic, but the models and tasks are somewhat dated, which weakens the generality and practical relevance of the conclusions.

- **Use more recent open-source LLMs (Critical):** The current experiments are based on GPT-2, Qwen2-0.5B, and Llama-2-7B. I would recommend adding experiments with newer small or medium-sized models to make the empirical conclusions more convincing and relevant to current LLM fine-tuning practice.

- **Evaluate on more realistic modern fine-tuning scenarios (Critical):** The current tasks, including sentiment classification, extractive question answering, and machine translation, are traditional NLP tasks and are closer to standard supervised fine-tuning benchmarks. They may not adequately represent modern LLM adaptation settings, such as instruction tuning, multi-turn dialogue fine-tuning, preference-related data, or tool-use/reasoning-oriented fine-tuning. Including more realistic modern fine-tuning scenario would substantially improve the paper’s relevance and generalizability.

- **Provide more fine-grained mechanistic analysis (Strengthen):** The paper would benefit from more fine-grained conclusions beyond the current layer-level observations. For example, regarding the conclusion that noise mainly affects task-specific middle-to-later layers, it would be useful to further analyze the attention patterns of different heads, identify which heads or representations are more task-relevant, and study how different types of noise affect these components. Such analyses could provide more actionable insights, for example for head-level fine-tuning, representation transfer, or targeted robust fine-tuning methods. More fine-grained analyses of this kind would make the work more useful and could bring new insights to the field.

---

> ### Author Response · Authors · 2026-06-17
>
> Thank you for you detailed comments. We have considered all your suggestions and submitted a revised draft. Our response to your concerns is as follows.
>
> **Use of more recent LLMs** To address this issue, we have now added Qwen2.5-1.5B-Instruct  model to our experiment setup.
>
> **Evaluate on more realistic modern fine-tuning scenarios** To address this issue we have now added one Math reasoning task for all the tasks.
>
> **Fine-grained mechanistic analysis** We have now added an attention-head-level analysis in Figure 7 and Figure 8. Specifically, we identify the top-3 attention heads that change the most in response to different types of noise. We observe that, for each task and model, there is substantial overlap across noise types, with two out of the three most affected heads often being shared. This suggests that each model contains certain task-specific attention heads that are particularly sensitive to noise.
>
> Please let us know if you have any further concerns.

---

> > ### Author Response · Authors · 2026-06-22
> >
> > To further improve the generalisability of the results, we have also included Gemma3-1B-Instruct model in our experimental setup. Please let us know if you need any further clarification.

---

### Review · Reviewer_VUSN · 2026-06-10

**Summary Of Contributions:**

This paper systematically studies how three types of training-data noise, label flip, typographical noise, and grammatical noise, affect three pretrained model families (GPT-2, Qwen2-0.5B, Llama-2-7B) on three NLP tasks (sentiment classification, question answering, machine translation). The paper analyzes how the noise sources affect model behavior on both prediction and representation levels. The paper analyzes how the noise sources affect model behavior on both prediction and representation levels. The results show that label noise is most harmful, while grammatical and typographical noise can be neutral or mildly beneficial; noise effects are localized to task-specific later layers, and attention is comparatively stable.

Pros:
1. The paper is well written and easy to follow.
2. The experimental design is thorough. It covers three model families, three NLP tasks, three noise types, and three corruption rates.
3. The representation level evaluation metrics are well chosen and complement each other.
4. The Section 6 robust vs. vulnerable stratification shows that representation drift can happen even when predictions do not change, which is an interesting observation.

Cons:
1. Several citations are unresolved BibTeX keys like liu2020early, tanzer2022memorisation, chen2025basin, etc.
2. There are some mismatches between the code and the paper, especially in hyperparameter settings. The authors should verify that every setting is consistent with what was actually run for the reported results.

**Audience:**

Yes

**Audience Explanation:**

This work addresses a gap in LLM noise-robustness research and is of interest to the TMLR audience, with practical implications for robust fine-tuning.

**Claims And Evidence:**

Yes

**Claims Explanation:**

The claims are well supported by systematic experiments across three model families on three NLP tasks. The same qualitative findings appear under multiple analyses. The agreement across these independent diagnostics is convincing. Some minor gaps remain that may need further investigation.

**Requested Changes:**

1. Check and fix the unresolved citations.

2. Does Table 2 report single-seed or multi-seed results? Based on Appendix F's reference to the "default seed 42," it appears to be single-seed. If so, multi-seed numbers should be reported in the main table.

3. Reconcile the hyperparameters and settings between the paper (Tables 8 and 9, Section 4) and the released code, including training set sizes, learning rates, test sets, and decoding strategy, etc.

4. The probing methodology in Section 3.2 could be more detailed. The released code suggests several design choices that are not stated in the paper, and should be stated explicitly.

5. In Table 2, why does Llama-2 SC at Label Flip 30% (95.12) score higher than the clean baseline (94.12)? Also, all TN and GN rates on Qwen2 QA beat the clean baseline of 68.00. Is there any explanation for these observations?

6. Section 6 finds that representation drift can happen even when predictions do not change. Currently only label-flip noise is analyzed (Appendix D), could this analysis be extended to TN and GN?

Minor:
1. Footnote 2 cites Wikipedia for the Frobenius norm, please use a textbook reference.
2. P8 Section 5.1, the amount of 'noise' caused by label noise, should be 'damage'?
3. P10 Section 5.3, level flip -> label flip

---

> ### Author Response · Authors · 2026-06-17
>
> Thank you for your comments. We have considered all your suggestions and submitted a revised draft. Our response to your concerns is as follows.
>
> **Unresolved Citations** We have now fixed all the unresolved citations in the updated draft.
>
> **Multi-Seed Experiments**  We have now added multi-seed results in Table 2. We have reported the mean and standard deviation across multiple seeds.
>
> **Hyperparameter Setting and Decoding Strategy**  We have added all the hyperparameter details and decoding setup in Appendix B and C, and made them consistent in the paper and the code.
>
> **Probing Strategy Setup** Based on Reviewer 1's suggestion we have now updated the probing strategy for question answering, machine translation and math reasoning. We have updated the methodology in Section 3.2. The code for probing is also now updated.
>
> **Llama-2 SC at Label Flip 30\% (95.12) score higher than the clean baseline (94.12)**  Similarly to existing work [1] our explanation for this behavior is that a small amount of noise can sometimes help to increase the robustness of the model. Hence the performance of label flip noise is more than its original version. Similar observation can be found for Qwen model for typo and grammatical noise.
>
> [1] "Data Noising as Smoothing in Neural Network Language Models"-ICLR 2017
>
> **Section 6 Representational Drift only on Label Flip**  We have now added the results for typo and grammatical noise Figure 13 in Appendix.
>
> We have also fixed all the typos mentioned in your minor comments. We also fixed the Frobenius norm citation in the updated draft. Please let us know if you have any further concerns.

---

### Author Response · Authors · 2026-06-17

We would like to thank all the reviewers for their comments. We have now updated the draft based on the suggestions made by reviewers. The overall changes in the revised version are as follows.

**New Model & Task** We have now added Qwen2.5-1.5B Instruct model and Gemma3-1B and Math reasoning task in all of our experiments.

**Clean vs Clean and Noisy vs. Noisy Baseline** We have now added baselines on Noisy vs. noisy and  clean vs. clean in  our experiments.

**Multi-Seed Experiments** We have now added multi-seed results with mean and variation in Table 2.

**Finegrained Mechanistic Analysis** We have now added attention head based finegrained mechanistic analysis in all of our experiments.